# An Interactive Paradigm for Deep Research

**Lin Ai** [1 2]  **Victor S. Bursztyn** [2]  **Xiang Chen** [2]  **Julia Hirschberg** [1]  **Saayan Mitra** [2]

## Abstract

Recent advances in large language models (LLMs) have enabled deep research systems that synthesize comprehensive, report-style answers to open-ended queries by combining retrieval, reasoning, and generation. Yet most frameworks rely on rigid workflows with one-shot scoping and long autonomous runs, offering little room for course correction if user intent shifts mid-process. We present STEER, a framework for **St**eerable d**eE**p **R**esearch that introduces interpretable, mid-process control into long-horizon research workflows. At each decision point, STEER uses a cost–benefit formulation to determine whether to pause for user input or to proceed autonomously. It combines diversity-aware planning with utility signals that reward alignment, novelty, and coverage, and maintains a live persona model that evolves throughout the session. STEER outperforms state-of-the-art open-source and proprietary baselines by up to 22.80% on alignment, leads on quality metrics such as breadth and balance, and is preferred by human readers in 85%+ of pairwise alignment judgments. We also introduce a persona–query benchmark and data-generation pipeline. To our knowledge, this is the first work to advance deep research with an interactive, interpretable control paradigm, paving the way for controllable, user-aligned agents in long-form tasks.

## 1. Introduction

Recent advances in large language models (LLMs) have shifted information access from ranked retrieval to systems that generate comprehensive, report-style answers to complex and open-ended queries (Du et al., 2025). These *deep research* systems, spanning proprietary platforms (Google,

2024; OpenAI, 2025; xAI, 2025) and open-source frameworks (Elovic, 2025; LangChain, 2025), combine iterative retrieval with multi-step reasoning to synthesize well-supported outputs (Coelho et al., 2025).Benchmarks such as DeepResearchGym (Coelho et al., 2025) and DeepResearch Bench (Du et al., 2025) have begun to standardize this setting, providing realistic research-style questions and automated evaluation protocols for long-form, citation-heavy reports. However, these benchmarks evaluate research-style queries and long-form reports against generic quality criteria; they do not provide *persona-conditioned, aspect-grounded targets* needed to measure alignment and focus when a system is meant to adapt to an individual user. This leaves a gap for evaluating interactive, user-steerable research agents, which we address with a persona-query suite that pairs deep-research-worthy queries with query-conditioned personas and actionable aspect checklists. On the system side, current deep research agents largely fall into two paradigms: multi-agent pipelines that divide planning, search, and synthesis (Huang et al., 2025; Alzubi et al., 2025; Li et al., 2025a; Zhang et al., 2025a), and RL-trained agents that learn to search and reason effectively (Zheng et al., 2025b; Jin et al., 2025; Song et al., 2025). Yet, regardless of architecture, most systems follow a rigid workflow: one-time scoping (often with a single clarification), followed by a long autonomous run. If user intent shifts mid-process, there is little room to course-correct, resulting in wasted cost and misaligned reports. This highlights the need for an alternative design where mid-process interaction is central, not optional.

Two research threads closely relate to our work. *Personalization and alignment* examine how to tailor LLM outputs to user intent, from profile-conditioned generation (Wu et al., 2025) to long-form checklists (Salemi et al., 2024; Salemi & Zamani, 2025) and interactive preference elicitation. While these works show the value of personalization, most assume fixed personas or separate preference modeling from system control, lacking a principled way to determine when to seek input. *Interactive reasoning* investigates how LLMs ask clarifying questions (Andukuri et al.; Ren et al., 2023; Wu et al., 2024), model future turns (Zhang et al., 2025b), or learn clarification policies (Chen et al., 2025). These methods address a *local clarification* bottleneck — detecting ambiguity in the current request and deciding whether

[1]Department of Computer Science, Columbia University, New York, NY, USA [2]Adobe Research, San Jose, CA, USA. Correspondence to: Lin Ai <lin.ai@cs.columbia.edu>.

*Proceedings of the 43rd International Conference on Machine Learning*, Seoul, South Korea. PMLR 306, 2026. Copyright 2026 by the author(s).

| System | Mid-process steering | Adaptive pause decision | Live persona modeling |
|---|:---:|:---:|:---:|
| *Deep Research Framework (Open-Source)* | | | |
| **DeepResearcher** (Zheng et al., 2025b) | ✗ | ✗ | ✗ |
| **Search-r1** (Jin et al., 2025) | ✗ | ✗ | ✗ |
| **ManuSearch** (Huang et al., 2025) | ✗ | ✗ | ✗ |
| **Search-o1** (Li et al., 2025a) | ✗ | ✗ | ✗ |
| **GPT-Researcher** (Elovic, 2025) | ✗ | ✗ | ✗ |
| **Open Deep Research** (LangChain, 2025) | ◆ | ✗ | ✗ |
| **ERD** (Prabhakar et al., 2025) | ✓ | ✗ | ✗ |
| *Deep Research Framework (Proprietary Web-Based)* | | | |
| **OpenAI Deep Research** (OpenAI, 2025) | ◆ | ✗ | ✗ |
| **Google Gemini Deep Research** (Google, 2024) | ◆ | ✗ | ✗ |
| *Interactive/Persona-Aware Reasoning Frameworks* | | | |
| **PersonaAgent** (Zhang et al., 2025c) | ✗ | ✗ | ✓ |
| **ReasonGraph** (Li et al., 2025b) | ✓ | ✗ | ✗ |
| **HITL CoT MCS** (Cai et al., 2023) | ✓ | ✗ | ✗ |
| **STEER** | ✓ | ✓ | ✓ |

*Table 1.* Comparison of deep research frameworks (open-source and proprietary) and interactive/persona-aware reasoning frameworks in terms of mid-process steering via adaptive pause decisions and live persona modeling. To our knowledge, STEER is the first benchmarkable deep-research framework that jointly offers all.

to answer it directly or ask first — rather than an end-to-end policy over where to pause in a long-horizon trajectory, how pausing trades off against exploration, and how user goals should evolve mid-process. A complementary thread exposes reasoning traces for human inspection: INTERACTIVE REASONING (Pang et al., 2025) renders chain-of-thought as an editable topic hierarchy, and REASONGRAPH (Li et al., 2025c) visualizes sequential or tree-based reasoning paths. These tools improve transparency and oversight, but provide *static visualization* rather than an adaptive pause mechanism or live persona model that conditions planning, branch utility, and synthesis within a single long-horizon research tree. Existing approaches thus either optimize autonomous agents or isolate clarification as a narrow skill. In contrast, we aim to offer an integrated control paradigm that jointly governs pausing, exploration, and personalization mid-process.

We introduce STEER, a framework for **St**eerable d**eE**p **R**esearch that brings interactive control to long-horizon research workflows (Figure 1). The key intuition is that deep research should occasionally *ask*, not just *answer*: STEER uses a cost–benefit formulation at each decision point to determine whether to pause for user input or to proceed autonomously. To remain both user-aligned and exploratory, it combines diversity-aware planning with utility signals that reward alignment, novelty, and coverage. A live persona is continuously updated based on interactions and conditions all downstream planning, scoring, and synthesis, enabling the system to adjust as user needs evolve.

Table 1 situates STEER among deep research frameworks and interactive/persona-aware reasoning systems. Most deep research systems still do one-shot scoping followed by a long autonomous run: once a query is issued, a fixed pipeline executes to completion with no interpretable way

to intervene in the research tree or to decide when to solicit guidance (◆ denotes a single upfront scoping/plan-confirmation step). Concurrent work, EDR (Prabhakar et al., 2025), adds user-initiated mid-process steering via an exposed task plan, but lacks an adaptive pause policy and live persona modeling. Interactive/persona-aware systems address complementary aspects: they maintain evolving user representations or expose reasoning for inspection and manual correction, but they do not implement an adaptive pause policy that decides *where* to pause and *how* to balance autonomy and control. In contrast, STEER is, to our knowledge, the first benchmarkable deep research framework that jointly supports mid-process steering, an adaptive pause mechanism, and live persona modeling.

Our contributions are as follows:

- We propose STEER, an interactive deep research framework that supports interpretable, mid-process control and dynamic user alignment throughout the research loop.
- Extensive experiments show that STEER outperforms the strongest open-source and proprietary OpenAI baselines on persona-tailoredness and report quality, while offering fine-grained control to tune trade-offs between alignment and user burden, as well as between under-exploration and over-personalization. A human study further confirms its preference among readers, with significant gains in alignment, focus, and usability.
- We introduce a persona–query evaluation suite and a reusable data generation pipeline grounded in prior benchmarks, suitable for future evaluation and training of interactive deep research agents.

In summary, STEER consistently outperforms strong open and proprietary baselines, achieving 7.83%–22.80% higher alignment and leading on general quality metrics such as

breadth and balance. Human readers prefer STEER in over 85% of alignment and 83% of focus pairwise comparisons. To our knowledge, *this is the first work to* **advance deep research with an interactive, interpretable control paradigm**. We believe that this paradigm shift will shape future long-horizon research agents, enabling decision policies that adapt to individual users and their evolving needs, rather than relying on a single upfront clarification.

## 2. STEER

### 2.1. Problem Setup and Objectives

We formulate steerable deep research as an interactive planning task. Given a user query $Q$, the system incrementally constructs a research tree and produces a cited synthesis report $R$. The goal is to generate a report that is both high-quality and aligned with the evolving preferences of the user, while keeping the number of interruptions minimal and well-timed.

Each user is represented by a persona $P = (p_{\text{text}}, \mathcal{A})$, where $p_{\text{text}}$ is a natural-language description combining profile and personality traits (following Wu et al. (2025)), and $\mathcal{A}$ is a set of aspects the user expects to see addressed in the final report. We evaluate reports along two complementary dimensions: **(1) Alignment**: the extent to which the report covers the aspects in $\mathcal{A}$; and **(2) Focus**: the proportion of content that remains on-topic with respect to $\mathcal{A}$.

### 2.2. System Overview

Our framework STEER transforms monolithic deep-research pipelines into an interactive process. The system is structured around three core components: **diversity-aware exploration**, **pause decision**, and **persona modeling**. At a high level, the framework incrementally builds a research tree that represents possible exploration paths and selectively engages the user at key checkpoints.

We denote the research tree as $T = (N, E)$, where each node $n \in N$ represents a sub-problem query with partial research results and each edge $(n, n') \in E$ indicates a decomposition into sub-directions. The tree expands level by level up to a maximum depth $D$, a hyper-parameter controlling how many layers are explored before synthesis. At each step, the system operates at a **frontier node** $n^\star$ and performs the following actions (Figure 1):

1. **Diversity-aware exploration:** Generate candidate follow-up directions from $n^\star$ and select a diversified subset of size $K$ to serve as potential expansions (see the *Planning* panel in Figure 1).
2. **Pause decision and expansion:** Compute branch utilities, execution costs, and the expected gain of asking, and compare this to the pause cost. If a pause occurs,

present the diversified subset to the user and then expand the user-selected items together with any newly suggested directions. Otherwise, expand the system-proposed diversified subset directly. Sub-agents then perform retrieval and reasoning at each expanded child to produce node-level reports (see the *Pause Decision* panel in Figure 1).
3. **Persona modeling:** Update the inferred persona $\hat{P}$ with signals from the query, initial profile, and any user feedback gathered during pauses. The updated persona conditions planning, utility scoring, and synthesis in subsequent steps (see the *Persona Modeling* panel in Figure 1).

The process terminates once all nodes at depth $D$ have been expanded, at which point the accumulated node reports are aggregated into the final report $R$. This interactive loop enables reports that are better aligned with user goals while minimizing redundant or off-topic exploration.

### 2.3. Diversity-Aware Exploration

As described in Section 2.2, at each frontier node $n^\star$, the system generates a set of follow-up questions as potential next steps. To promote exploration and reduce redundancy, we explicitly prompt for **distinct facets** and include one **wild-card** direction (see Appendix K for prompt details). From this candidate set, we select a diversified subset of size $K$ to either present to the user (if a pause is triggered) or expand automatically.

To select this subset, we apply a greedy Maximal Marginal Relevance (MMR) strategy (Carbonell & Goldstein, 1998; Wang et al., 2025), which balances confidence scores with dissimilarity to previously chosen directions. MMR is particularly well-suited to our setting: it is simple, efficient, and interpretable, while effectively encouraging topical coverage across different aspects. In contrast, alternative diversity methods (e.g., clustering or determinantal point processes) introduce additional complexity and hyper-parameters without clear gains in this context. Appendix C provides the full algorithmic details.

### 2.4. Pause Decision and Expansion

After the proposal stage has produced a diversified set of candidates, the system must decide whether to involve the user or continue autonomously. Asking everywhere is undesirable: user tolerance for interruptions is limited and varies widely. Some users prefer high-level guidance while trusting the system to handle details; others are more detail-oriented but want control only in specific themes. Preferences also shift across the depth of the research tree and over time. A well-calibrated system must respect these preferences while steering the exploration toward the user's goals. Below, we present the pause decision mechanism in a top-down structure: we begin with the overall decision rule and then

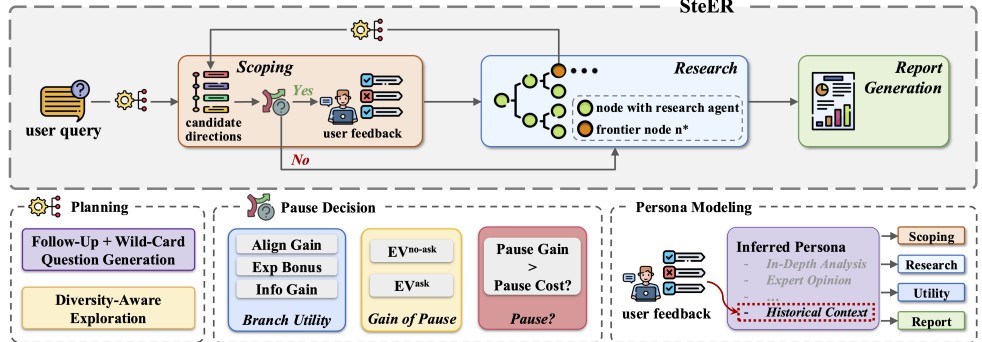

Figure 1. Overview of **STEER**. The upper panel shows the end-to-end pipeline. The lower panels zoom into the three core modules: *Planning*, *Pause Decision*, and *Persona Modeling*.

unpack its components, including pause cost, expected gain, and branch utility.

**Decision rule**   At each frontier node $n^\star$, the system evaluates whether pausing to ask the user is beneficial. This decision is framed as a cost–benefit comparison:

$$a(n^\star) = \begin{cases} \text{PAUSEASK}, & \Delta EV(n^\star) > C(n^\star), \\ \text{PROCEED}, & \text{otherwise.} \end{cases}$$

Here, $\Delta EV(n^\star)$ denotes the expected utility gain from pausing — by allowing the user to refine or redirect the next steps — while $C(n^\star)$ denotes the cost of interruption, scaled by user-specific tolerance.

**Pause cost**   Not all users interact in the same way. To model this, we assume two things: *(1)* a user's tolerance for interruptions decreases over time, and *(2)* users differ in how much interruption they are willing to tolerate in total, and how fast that tolerance depletes.

To capture this, we introduce two hyper-parameters:

- $C_0 \in [0, 1]$: the *base pause cost*. This reflects a user's general sensitivity to interruptions. A lower $C_0$ implies the user is open to frequent interaction; a higher $C_0$ indicates a preference for minimal disruption.

- $\text{Tol} \in \mathbb{N}$: the *tolerance budget*. This governs how quickly the pause cost increases with the number of questions asked. Intuitively, $\text{Tol}$ represents the approximate number of clarification questions the user is comfortable answering across the entire session.

A user may tolerate multiple clarifications within a single topic but become frustrated by interruptions scattered across too many unrelated ones. To reflect this, we distribute the global tolerance budget $\text{Tol}$ across all active *top-level directions*, defined as the root's immediate children. While users may have different preferences across themes, we simplify

by evenly dividing the tolerance budget across top-level directions. Each node $n$ belongs to a top-level direction $j \in K'$, where $K'$ denotes the number of currently active directions. If the system proceeds automatically, $K' = K$ (the full diversified set). If a pause occurs, $K'$ equals the number of user-selected plus user-added directions. The pause cost at a frontier node $n^\star$ is then computed as:

$$C(n^\star) = C_0 \cdot \left(1 + \frac{\text{pauses}_j}{\text{Tol}_j}\right),$$

where $\text{pauses}_j$ is the number of times the system has previously paused in direction $j$. As the number of pauses grows within a direction, the cost of pausing again increases proportionally.

**Pause gain**   The *gain of pausing* should reflect two factors: the utility we forgo by pruning branches and the execution cost we save by not pursuing them.

Let $\{n_k^\star\}_{k=1}^K$ be the candidate children at the frontier node, with branch utilities $U(n_k^\star)$ and normalized execution costs $C^{\text{exec}}(n_k^\star)$. If we proceed automatically, we pursue all $K$, and the expected value of the frontier node without pausing is $EV^{\text{no-ask}}(n^\star) = \sum_{k=1}^K U(n_k^\star) - \sum_{k=1}^K C^{\text{exec}}(n_k^\star)$. If we pause, the user keeps a subset $S \subseteq \{1, \ldots, K\}$, so the expected value of the frontier node with pause is $EV^{\text{ask}}(n^\star) = \sum_{k \in S} U(n_k^\star) - \sum_{k \in S} C^{\text{exec}}(n_k^\star)$. To estimate $S$, we retain candidates whose upper utility bound overlaps the leader's lower bound, capturing all options that are plausibly optimal. Equivalently, this decision rule prunes all branches whose best-case utility still falls below the worst-case value of the current leader. See Appendix D for bound construction and filtering.

A pause only changes which branches we do *not* execute. The gain of pausing at the frontier node is the saved cost

minus the lost utility of those pruned branches:

$$\Delta EV(n^\star) = EV^{\text{ask}}(n^\star) - EV^{\text{no-ask}}(n^\star)$$
$$= \sum_{k \in S^c} \left( -U(n_k^\star) + C^{\text{exec}}(n_k^\star) \right).$$

**Branch utility.** We score each candidate child $n_k^\star$ using a weighted combination of three factors:

$$U(n_k^\star) = \Delta\text{Align}(n_k^\star) + \lambda_{\text{exp}}\,\text{Explore}(n_k^\star)$$
$$+ \lambda_{\text{info}}\,\text{InfoGain}(n_k^\star),$$

where each component is scaled to $[0, 1]$ for direct comparability with the pause cost. (See Appendix D for exact computations and normalization.)

- **Alignment gain** ($\Delta\text{Align}$) computes predicted increase in persona alignment relative to the parent under the current inferred aspects $\hat{A}_s$. It rewards branches that cover more of what the user actually cares about.
- **Exploration bonus** (Explore) adds a small reward for under-explored facets to discourage repeatedly selecting the same angle. We capture this "reward under-explored, penalize over-explored" behavior with a lightweight *count-based bonus over facet tags*, inspired by the Upper Confidence Bound (UCB) family (Auer, 2002; Auer et al., 2002; Li et al., 2010): rarely used facets receive larger bonuses, and the bonus decays naturally as they are chosen more frequently. We use UCB here as an *optimism-based exploration prior* rather than a classical stationary-bandit algorithm. The arms (facets) are non-stationary and conditioned on an evolving persona, so we do not claim standard UCB regret guarantees. Instead, Explore provides a simple, interpretable inductive bias against repeated collapse onto the same facet, complementing MMR (local diversity) and InfoGain (semantic novelty) by promoting longer-horizon facet coverage.
- **Information gain** (InfoGain) measures the content-level novelty of a candidate's expected evidence relative to accumulated learnings. While Explore encourages facet-level diversity, InfoGain focuses on semantic-level novelty, prioritizing branches that are more likely to yield genuinely new information from the web.

$\lambda_{\text{exp}}$ and $\lambda_{\text{info}}$ balance breadth and novelty against alignment. Both Explore and InfoGain complement the diversify-aware exploration described in Section 2.3: while the latter ensures that the *initial question set* spans distinct facets, it does not guarantee that the resulting content will be diverse. Explore and InfoGain help mitigate this by promoting long-term diversity at the facet and content levels, respectively. While our process uses a minimal three-factor utility for clarity and stability, the framework is easily extensible — additional criteria (e.g., risk, credibility) can be incorporated as needed.

**Execution cost.** $C^{\text{exec}}(n_k^\star)$ estimates remaining work if we expand $n_k^\star$. It is also normalized to $[0, 1]$ so it is commensurate with utilities. We approximate the cost by the tokens of a saturated subtree beneath $n_k^\star$, as tokens provide a consistent, model-agnostic proxy, and correlate with both latency and spend. See Appendix D for computation details.

## 2.5. Persona Modeling

Beyond deciding *when* to ask (Section 2.4), the system must also know *who* it is optimizing for. In deep research, users often do not know exactly what they want at the start. Their goals shift as they encounter new information, and partial results may reveal new priorities. Fixing a full persona upfront risks overfitting to stale assumptions or flooding the system with irrelevant detail. To address this, we maintain a *live* persona that evolves dynamically as the research progresses.

At each $n^\star$, **STEER** maintains an updated persona estimate $\hat{P}(n^\star) = \left( \hat{p}_{\text{text}}(n^\star),\ \hat{A}(n^\star) \right)$, where $\hat{p}_{\text{text}}(n^\star)$ captures the user's profile and $\hat{A}(n^\star)$ represents the current inferred set of aspects the user cares about. When a pause occurs, we update $\hat{P}(n^\star)$ based on user-selected directions and any new suggestions, and implicitly incorporate recent research findings. This evolving persona conditions all downstream modules: it guides research and follow-up question generation, shapes the branch utility score via alignment to $\hat{A}(n^\star)$ (decision), and steers final report synthesis. See Appendix K for full details on how the persona is inferred and updated using LLM prompts (*Persona Checklist Inference* and *Persona Modeling* prompts), and how the evolving $\hat{P}(n^\star)$ is used across the planning, research, and synthesis pipeline.

A live persona keeps the interaction tightly aligned with the user's current interests. It prevents drift caused by outdated assumptions, reduces unnecessary questions by filtering irrelevant directions, and adapts to new priorities that emerge during exploration.

## 3. Experiments

### 3.1. Experimental Setup

**Evaluation data** We synthesize query–persona pairs by adapting established datasets and methods, with light modifications to better suit our goals. We begin with 1k queries from the *Researchy Questions* dataset (Rosset et al., 2024), as used in *DeepResearchGym* (Coelho et al., 2025). For each query, we generate a plausible user persona $p_{\text{text}}$ by adapting the ALOE profile–personality paradigm (Wu et al., 2025): we seed from ALOE profiles and prompt an LLM to propose new profiles that would reasonably ask the given query. To ensure diversity, we apply SBERT-based filtering (Reimers & Gurevych, 2019) and keep only distinct, plausible personas, following prior work (Wu et al., 2025; Wang

et al., 2023).

Given each $p_{\text{text}}$, we generate 5–8 evaluation aspects $\mathcal{A}$ using prompts inspired by Salemi & Zamani (2025), following their checklist format to ensure that the aspects are actionable, measurable, and grounded in the persona. This enables robust alignment and focus evaluation, avoiding the ambiguity of more generic rubrics.

Compared to Wu et al. (2025) and Salemi & Zamani (2025), our adaptations are minimal but tailored to deep research: **(i)** persona generation is query-conditioned to ensure relevance, **(ii)** diversity filtering is stricter to avoid near-duplicates, and **(iii)** aspects are framed for long-form, cited outputs. We evaluate on a held-out set of 200 queries. Full details of data generation are in Appendix E.

***User Agent*** **simulation**  To enable scalable, repeatable evaluation, we simulate user interactions with a *User Agent* conditioned on the full persona $P = (p_{\text{text}}, \mathcal{A})$. The agent selects directions that best align with $\mathcal{A}$ and proposes a new follow-up when uncovered aspects remain, yielding realistic steering signals without human-in-the-loop variability. (See Appendix N for the full prompt.)

**Metrics**  We evaluate persona-tailored quality using two proposed metrics: **Alignment** and **Focus**, both judged by *gpt-4.1-mini* following *DeepResearchGym*. (Prompts used to obtain these metrics are listed in Appendix M.) We present the meta-evaluation results for the LLM judge in Appendix I.

- **Alignment:** Given aspect set $\mathcal{A}$ and report $R$, we compute: $\text{Align}(R, \mathcal{A}) = \frac{1}{2|\mathcal{A}|} \sum_{a \in \mathcal{A}} \text{align}(R, a)$, $\text{align}(R, a) \in \{0, 1, 2\}$. Here, 0 means that the aspect is not addressed, 1 means that it is partially addressed (e.g., mentioned or vaguely covered), and 2 means that it is fully addressed with sufficient detail and evidence, all scored by the LLM-judge. This gives an interpretable, per-aspect measure of user alignment.
- **Focus:** We extract a set of keypoints $\mathcal{KP}$ — short, evidence-bearing spans — from $R$ using an LLM, and then ask the judge whether each keypoint ($k \in \mathcal{KP}$) maps to at least one user aspect: $\text{Focus}_{\text{kp}}(R, \mathcal{A}) = \frac{1}{|\mathcal{KP}|} \sum_{k \in \mathcal{KP}} \mathbf{I}[\text{map}(k) \neq \varnothing]$. While alignment is akin to *recall* over aspects, focus acts as a form of *precision*, rewarding dense, on-target content.

In addition, we report *DeepResearchGym*'s quality metrics, including clarity, depth, breadth, and insight, to evaluate general writing quality beyond persona targeting.

**Baselines**  We compare **STeER** to two strong open-source frameworks: *GPT-Researcher* (Elovic, 2025) and *Open Deep Research* (LangChain, 2025), both evaluated as top-performing frameworks (Coelho et al., 2025). On the proprietary model side, we benchmark against OpenAI's `o4-mini-deep-research` model.

We compare systems under a controlled setting: for **STeER** and the open-source frameworks, all agents use GPT-4o, the research tree is fixed (depth 3, breadth 3), outputs share the same token cap, and the only variable is persona information. For fairness, all baselines are run under three input settings: (1) query only, (2) query + initial persona (first sentence of $p_{\text{text}}$), and (3) query + full persona. This allows us to assess how well each baseline adapts to different levels of user information. Note that **STeER** always operates with only the initial persona, and must infer preferences dynamically throughout the interaction.

### 3.2. How Much Does STeER Improve Persona-Tailored Quality?

From Table 2, we see that **STeER** achieves the strongest persona-tailored performance on both metrics across all systems (e.g. 7.83% higher alignment than the runner-up GPT-Researcher$_{\text{full-persona}}$), even though some of those baselines are given the full persona, while **STeER** only receives the first sentence. This highlights the effectiveness of **STeER**'s interactive pausing and live persona modeling, which enable accurate mid-process adaptation without relying on full upfront persona input. This has practical appeal: real-world deployments often face privacy constraints, onboarding friction, or noisy user profiles. **STeER**'s ability to achieve strong alignment under minimal initial input makes it more robust in such settings.

**STeER** also leads in breadth and balance, reflecting the role of **STeER**'s diversity-aware exploration and utility components, Explore and InfoGain, in promoting semantic novelty and facet diversity. **STeER** also significantly outperforms the open-source baselines in depth and insight, though it falls slightly short of the proprietary OpenAI model on these metrics.

### 3.3. How Does STeER Provide Interpretable Controls for Optimal Pausing?

As introduced in Section 2.4, **STeER** offers two interpretable knobs to control pausing behavior: the base pause cost $C_0$, which sets the system's aversion to interruptions, and the tolerance budget Tol, which controls how quickly pause cost grows within a top-level direction. In this study, we vary $C_0$ while keeping Tol = 3 fixed. This is because, in shallow trees (depth 3), the effect of Tol is limited. Tol is more impactful in long-horizon tasks where user fatigue may accumulate across levels. Conceptually, Tol captures a user-specific interaction limit. We tune $C_0$ to match the pause budget (Tol) while maximizing gain per pause.

| Metric → | Persona-Tailored | | Quality | | | | |
|---|---|---|---|---|---|---|---|
| System ↓ | Align | Focus$_{kp}$ | Clarity | Depth | Breadth | Insight | Balance |
| GPT-Researcher | 66.63 | 78.42 | **81.80** | 86.30 | 88.40 | 76.60 | 81.25 |
| GPT-Researcher$_{initial-persona}$ | 74.59 | 81.68 | 79.05 | 87.37 | 88.71 | 75.52 | 81.71 |
| GPT-Researcher$_{full-persona}$ | _79.48_ | 83.83 | 77.93 | 87.09 | _90.31_ | 79.05 | 82.58 |
| OpenDeepResearch | 62.74 | 83.72 | 74.90 | 82.40 | 88.85 | 68.39 | 81.25 |
| OpenDeepResearch$_{initial-persona}$ | 69.79 | 85.45 | 72.51 | 81.64 | 84.12 | 68.98 | 77.44 |
| OpenDeepResearch$_{full-persona}$ | 77.20 | _86.10_ | 74.02 | 83.42 | 87.62 | 73.18 | 79.44 |
| o4-mini-deep-research$_{initial-persona}$ | 72.73 | 86.09 | 75.76 | **89.10** | 89.51 | **86.74** | _82.76_ |
| o4-mini-deep-research$_{full-persona}$ | 75.72 | 86.02 | 75.54 | 87.19 | 87.36 | _85.01_ | 82.63 |
| **STEER** | **85.70** | **86.45** | _79.97_ | _88.67_ | **91.29** | 83.04 | **84.19** |

*Table 2.* Performance comparison between **STEER** and baselines. For **STEER**, we report performance at $C_0 = 0.7$ (see Section 3.3 for selection rationale).

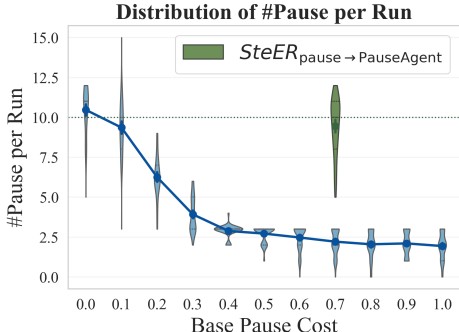

*Figure 2.* Distribution of number of pauses per run across base pause cost values.

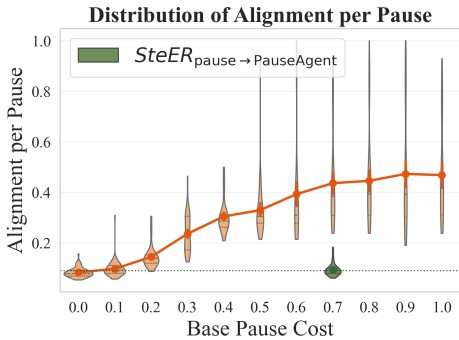

*Figure 3.* Alignment per pause across base pause cost values.

To benchmark against intuitive alternatives, we introduce a *PauseAgent* baseline that uses an LLM agent to predict pause vs. proceed at each frontier node (prompt in Appendix N). As shown in Figure 2 (and Appendix F.2), *PauseAgent* pauses excessively, far exceeding the Tol = 3 budget. In contrast, **STEER** with $C_0 \geq 0.4$ remains within budget, averaging fewer than 3 pauses.

Frequent pausing also hurts efficiency. Figure 3 shows that alignment per pause drops sharply at low $C_0$, while higher $C_0$ yields fewer but more impactful interventions. This trade-off is evident in Figure 4: while absolute alignment declines as $C_0$ increases, alignment and focus reach local

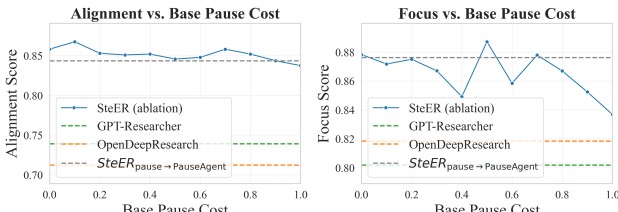

*Figure 4.* Effect of base pause cost on alignment (*left*) and focus (*right*). Baseline scores are shown as horizontal reference lines for comparison.

maxima around $C_0 = 0.7$, indicating a practical sweet spot.

In summary, **STEER** supports calibrated control of interaction. $C_0$ adjusts interruption cost directly, and Tol governs how that cost compounds over time. This formulation provides both interpretability and personalization, outperforming the *PauseAgent* baseline in effectiveness and flexibility.

### 3.4. How Does STEER Avoid Under-Exploration Driven by Personalization?

A potential failure mode is overfitting to personalization: when optimization focuses solely on aspect alignment, the system quickly collapses to a narrow trajectory, branch utilities flatten as $\Delta$Align approaches zero, and exploration stalls. To prevent this, **STEER** integrates three complementary signals at different axes.

First, diversity-aware exploration ensures that research directions span distinct facets at each step, avoiding early myopia. As shown in Table 3, removing it causes the largest drops in depth, breadth, and focus, along with a significant

| Method ↓ | Alignment | Focus$_{kp}$ | Depth | Breadth |
|---|---|---|---|---|
| **STEER** | **85.82** | **87.79** | 90.27 | **93.15** |
| (w/o) Explore | 84.98$_{\downarrow 0.98\%}$ | 85.17$_{\downarrow 2.98\%}$ | 89.86$_{\downarrow 0.45\%}$ | 92.60$_{\downarrow 0.59\%}$ |
| (w/o) InfoGain | 82.81$_{\downarrow 3.51\%}$ | 86.40$_{\downarrow 1.58\%}$ | 90.41$_{\uparrow 0.15\%}$ | 92.73$_{\downarrow 0.45\%}$ |
| (w/o) Div Explore | 84.57$_{\downarrow 1.46\%}$ | 84.29$_{\downarrow 3.99\%}$ | 88.63$_{\downarrow 1.82\%}$ | 91.09$_{\downarrow 2.21\%}$ |

*Table 3.* Ablation study on novelty and exploration components. Darker red indicates a larger performance drop relative to **STEER**.

decline in alignment, underscoring its role in maintaining structural and semantic diversity throughout the session.

In addition, two utility terms guide exploration: Explore encourages rotation across underrepresented facets, while InfoGain prioritizes semantic novelty. Ablating Explore leads to a large focus drop and notable declination in depth and breadth, with only a small impact on alignment, showing its importance in sustaining report-wide diversity. In contrast, removing InfoGain yields the largest alignment drop but only relatively modest effects on other metrics. This suggests that without semantic novelty, the system tends to dig deeper into already-favored lines, satisfying more user aspects while producing redundant evidence. These complementary behaviors introduce an interpretable trade-off: $\lambda_{\text{info}}$ prioritizes aspect satisfaction, while $\lambda_{\text{exp}}$ favors breadth; we set both to 0.5 for balance.

While our experiments focus on novelty and exploration, the utility function is extensible. Additional signals, such as factuality or plausibility, can be integrated into the same calibrated framework. Our contribution lies not in these specific factors, but in the interaction paradigm that supports modular, interpretable control over research behavior.

### 3.5. User Study Evaluation

To complement the automated LLM-judged metrics, we conducted a user study to evaluate whether **STEER** is preferred by human users. We compared **STEER** with GPT-Researcher and `o4-mini-deep-research` on 20 query–persona pairs. 12 annotators (all NLP/CS graduate students) viewed two reports for the same pair (one from **STEER**, one from a baseline) in randomized order on our custom annotation platform. Annotators saw only the two final reports, side by side in randomized order — no clarification questions, interaction traces, or intermediate system behaviors were exposed — and were unfamiliar with our system or the baselines' output formats. We did not observe consistent structural cues that would make a system identifiable (e.g., tables of contents appeared across systems). We nonetheless acknowledge that perfect blinding is difficult in any open-ended generation setting; see Appendix G and Limitations (Appendix J) for further discussion. For

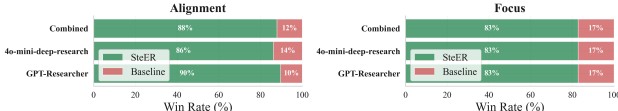

Figure 5. Pairwise human preference win rates on *Alignment* and *Focus*.

each comparison, annotators judged ***Alignment*** (better coverage of persona aspects), ***Focus*** (more on-topic with less redundancy), ***Coverage*** (aspect-level 0–2, averaged), and ***Findability*** (report-level 0–2 for ease of locating relevant information).

This design captures both quality and usability: *Alignment* and *Focus* reflect perceived persona-fit[1]; *Coverage* measures how thoroughly user interests are addressed, and *Findability* assesses how easily users can locate what matters. Full platform design and annotator instructions are in Appendix G.

We collected 58 valid pairwise annotations. To validate the quality of the annotations, we computed agreement using pairwise metrics across all annotator pairs for each evaluation dimension. Specifically, we report raw pairwise agreement and Gwet's AC1 (Gwet, 2008), a prevalence-resistant chance-corrected agreement coefficient that avoids the artificial deflation often observed with Fleiss' $\kappa$ (Fleiss, 1971) when the label distribution is skewed. As summarized in Table 4, for *Alignment* and *Focus* pairwise preferences, annotators achieve raw pairwise agreement of 82.0% and 73.8%, with AC1 values of 0.639 and 0.475, respectively, indicating substantial and moderate agreement. For *Coverage* and *Findability*, we observe similar patterns of fair to moderate agreement. For *Coverage*, raw agreement is 65.9% for **STEER** and 65.2% when aggregating baselines (GPT-Researcher and `o4-mini-deep-research`). For *Findability*, raw agreement is 75.4% for **STEER** and 65.6% for the aggregated baselines. These values reflect consistent, non-trivial consensus across annotators on all dimensions, especially given the inherent subjectivity of report quality.

As shown in Figure 5, **STEER** is preferred in about 86–90% of cases for Alignment and about 83% for Focus across GPT-Researcher and `o4-mini-deep-research`.

Figure 6 shows significant gains in *Coverage* and *Findability* for **STEER**. On a 0–2 aspect-coverage scale, **STEER** improves the average by +0.623 (from 0.828 to 1.451, $p = 3.05e - 12$), a relative improvement of about 75% which indicates a shift from below "somewhat covered" toward between "somewhat" and "fully" covered. On the 0–2 Findability scale, **STEER** improves by +0.690 (from 0.845 to 1.534, $p = 1.64e - 11$), moving readers from mostly

| Metric | Raw Agreement (%) | Gwet's AC1 |
|---|---|---|
| ***Alignment*** | 82.0 | 0.639 |
| ***Focus*** | 73.8 | 0.475 |
| ***Coverage*** | | |
| SteER | 65.9 | 0.318 |
| Baselines | 65.2 | 0.303 |
| ***Findability*** | | |
| SteER | 75.4 | 0.508 |
| Baselines | 65.6 | 0.311 |

*Table 4.* Inter-annotator agreement for user study annotations.

---

[1]Note that the *Alignment* and *Focus* metrics used in the user study are based on pairwise human preferences and are not directly comparable to the automatic metrics defined in Section 3.1.

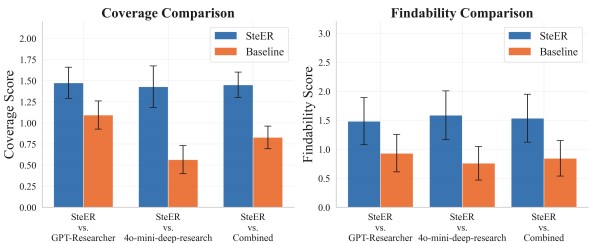

*Figure 6.* Human ratings on *Coverage* and *Findability*. *Left:* Average aspect-level *Coverage* scores of **STEER** and baselines. *Right:* Average *Findability* scores of **STEER** and baselines.

difficult-to-medium retrieval to comfortably above medium and closer to "easy to find." Together, these results indicate that **STEER** produces reports that are both better aligned with persona needs and easier to navigate.

## 4. Discussion

***User Agent* simulation**   To understand how our pause policy translates into user-facing behavior, we analyzed the *User Agent* used in offline evaluation (Appendix F.4). The *User Agent* maintains very high precision across base-pause costs ($> 0.97$), while recall declines as $C_0$ increases, and report alignment closely tracks *User Agent* recall ($r \approx 0.81$). This indicates that pausing affects outcomes primarily by changing how many promising directions are retained and developed, offering a controllable alignment–effort frontier via $C_0$. We view the *User Agent* as a diagnostic tool for sweeping policies and stress-testing settings, but acknowledge that real users may be noisier and value exploration differently; future work will calibrate the *User Agent* with human logs and run counterfactual replays to quantify gaps between simulated and actual behavior.

**Persona modeling**   We also examine how well **STEER**'s live persona tracks report quality. A useful takeaway from Appendix F.3 is that **STEER** not only pauses effectively but also recovers and maintains an accurate persona during a run. Even with only the first persona sentence as input, the inferred persona's alignment with the ground-truth aspect set strongly tracks final report alignment ($r \approx 0.85$, $p < 10^{-3}$), indicating that the learned persona is informative rather than decorative. As $C_0$ increases, pauses become fewer, the inferred persona is less specified, and downstream alignment declines. In practice, persona–report agreement is a useful diagnostic for selecting $C_0$: choosing the smallest $C_0$ that achieves a target agreement while balancing the alignment–effort trade-off.

**Broader application**   Beyond our experiments, **STEER** suggests a general pattern for long-horizon, high-stakes tasks that must balance personalization with exploration un-

der interpretable control. For instance, scientific-discovery agents and research stacks could benefit from pausing and live-persona steering to curb drift while preserving exploration (Team et al., 2025; Schmidgall & Moor, 2025; Zheng et al., 2025a). Likewise, high-stakes domains such as financial advising and trading (Zhang et al., 2024; Yu et al., 2024) and law and policy research (Li et al., 2024; Pipitone & Alami, 2024) are natural application areas for **STEER**'s interpretable, user-steerable control. Because of **STEER**'s modularity, domains can add factors such as factuality, citation quality, or safety alongside novelty and exploration. We view validating these extensions as promising future work.

## 5. Conclusion and Future Work

We have presented **STEER**, proposing a new *interactive paradigm* for deep research. **STEER** couples a cost–benefit pause policy with interpretable controls, a live persona that adapts mid-process, and diversity–novelty utility signals that keep exploration purposeful. Our experiments show that **STEER** improves persona-tailored quality by 7.83%–22.80% over strong open-source and proprietary systems, leads on generic quality metrics, and is preferred by human readers in over $85\%$ of alignment and $83\%$ of focus pairwise judgments. We also release a persona–query evaluation suite and data pipeline to support reproducible testing and future model development.

Looking ahead, several directions appear especially promising. On the system side, exploring speculative pre-execution to reduce latency, a dynamic breadth–depth planner, and policy learning for pause and branch selection could further strengthen real-time usability. On the evaluation side, end-to-end user studies evaluating full interactions, including task success, time to insight, perceived control and trust, and cognitive load, would better capture real-world value. We also see studying **STEER**'s robustness to noisier real-user behavior — contradictory feedback, partial or incomplete responses, and outright abandonment of the clarification process — as an important next step, naturally tied to preference-conflict handling in the live persona model (see Appendix J).

## Impact Statement

This work presents **STEER**, a framework for interactive, human-guided deep research agents. Our primary contribution is moving away from monolithic autonomous workflows toward a paradigm where model reasoning is transparent and steerable by the user.

This research has the potential to significantly enhance productivity in knowledge-intensive sectors by reducing the cognitive load of information synthesis. By enabling agents to adapt to user intent in real-time, we democratize access to

complex research capabilities, allowing non-experts to navigate dense technical or domain-specific literature effectively. Furthermore, our proposed adaptive pause mechanism improves computational efficiency by preventing agents from wasting resources on misaligned or irrelevant search trajectories.

We acknowledge two primary risks associated with highly personalized research agents:

- **Automation Bias and Over-Reliance:** As agents become more capable of long-horizon reasoning, users may uncritically accept generated summaries. **STEER** mitigates this by design; the interactive nature of the framework forces periodic user engagement, keeping the human in the loop and encouraging critical evaluation of the agent's intermediate findings rather than passive consumption of a final output.

- **Echo Chambers and Confirmation Bias:** Our system's live persona modeling optimizes for user alignment, which carries the risk of reinforcing existing biases by prioritizing information that aligns with the user's preconceptions. To address this, future deployment of such systems should include diversity-aware exploration in the frontier selection algorithm (as described in Section 2.3 to ensure opposing viewpoints are surfaced, even when they conflict with the modeled persona.

Ultimately, this work aims to align powerful autonomous agents with human values. By prioritizing steerability over pure autonomy, we provide a blueprint for safer AI systems that remain responsive to human oversight during complex tasks.

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

## A. Related Work

**Deep research** LLM-based research agents combine retrieval and multi-step reasoning to produce long-form answers (Coelho et al., 2025). Among the open-source frameworks, two dominant paradigms are multi-agent pipelines that split planning, browsing, and reading across roles (Huang et al., 2025; Alzubi et al., 2025; Li et al., 2025a; Zhang et al., 2025a) and RL-trained agents that learn to search and reason (Zheng et al., 2025b; Jin et al., 2025; Song et al., 2025). On the evaluation side, benchmarks

such as DeepResearchGym (Coelho et al., 2025) and Deep-Research Bench (Du et al., 2025) have begun to standardize this setting, providing realistic research-style questions and automated evaluation protocols for long-form, citation-heavy reports. Despite progress, most systems still follow one-shot scoping with at most a single clarification, then a long autonomous run and a monolithic report, offering little mid-process control when user needs evolve. Concurrent work, EDR (Prabhakar et al., 2025), adds user-initiated mid-process steering via an exposed task plan, but lacks an adaptive pause policy and live persona modeling.

**Personalization and alignment** A growing line of work pursues personalization for LLM agents, moving from static profile–personality conditioning and long-form checklists (Wu et al., 2025; Salemi et al., 2024; Salemi & Zamani, 2025) toward interactive, test-time adaptation and multi-stakeholder alignment (Xie et al., 2025). Recent trends probe persona behavior in interaction (e.g., consistency and drift under dialogue) (Frisch & Giulianelli, 2024) and build agent mechanisms that adapt actions to user preferences at inference time (Zhang et al., 2025c). While this work establishes that preferences should be updated during use, most approaches still lack *interpretable, end-to-end controls* for deciding *when* to seek input and *how* to steer long-horizon generation as goals evolve.

**Interactive reasoning and control** Another closely related line of work equips LLMs with interactive reasoning via clarification. Prior studies train models to ask when information is missing (Andukuri et al.; Ren et al., 2023; Wu et al., 2024), model future turns to decide ask vs. answer (Zhang et al., 2025b), and learn clarification policies with contrastive objectives (Chen et al., 2025). Visualization tools improve transparency and user steering over chains of thought (Pang et al., 2025; Li et al., 2025c). However, these efforts mostly address local interactions or static control, rather than providing interpretable, end-to-end controls for *when* to pause, *what* to explore, and *how* to adapt personalization mid-process in long-horizon research.

Table 1 positions STEER against both deep research frameworks and interactive/persona-aware reasoning systems, clarifying which capabilities each class actually offers. Open-source deep research frameworks and proprietary web-based services are all built around long autonomous run paradigm: once the user issues a query, a fixed pipeline executes to completion with no exposed, interpretable control over where in the research tree to intervene or when to ask for guidance. The ◆ marks in the mid-process steering column indicate the limited behavior: these systems sometimes allow a single upfront scoping or plan-confirmation step before the full autonomous run, but provide no further steering within the trajectory; beyond that point they are fully autonomous. In addition, for proprietary systems, public documentation suggests that fixed personas or user memory may be used, but there is no evidence of live persona modeling, and their control policies and internal mechanisms are not accessible or benchmarkable; we therefore restrict ourselves to conceptual comparison.

Meanwhile, interactive/persona-aware frameworks occupy the complementary side of the space: PersonaAgent (Zhang et al., 2025c) maintains a live user representation and adapts over time but has no steering capability, while ReasonGraph (Li et al., 2025b) and HITL CoT MCS (Cai et al., 2023) expose mid-process reasoning for visualization, inspection, or human correction but implement no adaptive pause mechanism and do not perform live persona modeling. Taken together, the table shows that existing systems provide at most one of the three capabilities we target — mid-process steering, adaptive pause decisions, or live persona modeling — and never all three in a benchmarkable deep research setting. **STEER** is the only system that supports all three simultaneously, coupling a cost–benefit pause policy with a live persona that conditions planning, branch utility, and synthesis within a single research tree.

# B. Use of LLMs for Writing Assistance

We used ChatGPT-4o *only* for language-level editing. Concretely:

- Polishing prose, tightening sentences, fixing grammar and LaTeX wording, reordering or shortening paragraphs, and suggesting alternative titles or section headers.
- No ideas, methods, claims, proofs, experiments, numbers, figures, tables, code, prompts, or citations were produced by the model. All technical content, analyses, and results were authored and verified by the authors.
- We supplied already written passages or outlines and requested editing (for example, "polish wording, keep all technical details unchanged").
- The model was not given proprietary data, code, or unpublished results beyond the text to be edited. All outputs were reviewed by the authors for accuracy and tone.

# C. Diversified Subset Selection

For completeness, we include the pseudocode of the greedy MMR selection used in our framework. Given a candidate set of follow-up questions $\mathcal{C} = \{q_1, \ldots, q_M\}$ with confidence scores $\text{conf}(q_i)$ and embeddings $\mathbf{e}_i$, the algorithm selects a diversified subset $\mathcal{C}'$ of size $K$:

**Require:** Candidate list $\mathcal{C} = \{q_1, \ldots, q_M\}$ with confidences $\mathrm{conf}(q_i)$, embeddings $\mathbf{e}_i$; desired subset size $K$
**Ensure:** Diversified subset $\mathcal{C}'$
1: $\mathcal{C} \leftarrow$ sort $\mathcal{C}$ in non-increasing order of $\mathrm{conf}(q_i)$
2: $\mathcal{C}' \leftarrow \emptyset$, $I_{\mathcal{C}'} \leftarrow \emptyset$
3: **while** $|\mathcal{C}'| < K$ **do**
4: $\quad C \leftarrow \{i \mid i \notin I_{\mathcal{C}'}\}$
5: $\quad$ **if** $I_{\mathcal{C}'} = \emptyset$ **then**
6: $\quad\quad i^\star \leftarrow \min C$ {top-confidence question}
7: $\quad$ **else**
8: $\quad\quad$ **for** $i \in C$ **do**
9: $\quad\quad\quad d_i \leftarrow \max_{j \in I_{\mathcal{C}'}} \mathrm{sim}(\mathbf{e}_i, \mathbf{e}_j) + \varepsilon$
10: $\quad\quad$ **end for**
11: $\quad\quad i^\star \leftarrow \arg\min_{i \in C} d_i$ {least similar to current set (MMR criterion)}
12: $\quad$ **end if**
13: $\quad \mathcal{C}' \leftarrow \mathcal{C}' \cup \{q_{i^\star}\}, \quad I_{\mathcal{C}'} \leftarrow I_{\mathcal{C}'} \cup \{i^\star\}$
14: **end while**
15: **return** $\mathcal{C}'$

| Data Split $\rightarrow$ | All | Eval |
|---|---|---|
| Total Queries | 1000 | 200 |
| Total Query-Persona Pairs | 1381 | 286 |
| Queries with 1 Persona | 646(64.6%) | 125(62.5%) |
| Queries with 2 Personas | 327(32.7%) | 64(32.0%) |
| Queries with 3 Personas | 27(2.7%) | 11(5.5%) |

*Table 5.* Data Statistics

## D. Details for Gain of Pausing Implementation

**Alignment gain** Let $r(n)$ denote the chunk report at node $n$, formed by concatenating the learnings $\{\ell_i\}_{i=1}^m$ (if there are $m$ learnings at the node), and let $\hat{A}_n$ be the inferred aspect set at that node. For the $k$-th child node of a frontier node $n_k^\star$,

$$\Delta\mathrm{Align}(n_k^\star) = \mathrm{Align}\big(r(n_k^\star), \hat{A}_{n^\star}\big) \\ - \mathrm{Align}\big(r(n^\star), \hat{A}_{n^\star}\big).$$

**Exploration bonus** For each chunk report, we prompt an LLM to assign short tags (see Appendix K, *Search Result Processing* prompt). We maintain the global tag set $\mathcal{T}$ and a cumulative usage count $\mathrm{count}(T)$ for each tag $T$ up to the current step. With a small constant $\epsilon > 0$, the exploration bonus is

$$\mathrm{Explore}(n_k^\star) = \frac{1}{|\mathcal{T}|} \sum_{T \in \mathcal{T}} \frac{\epsilon}{1 + \sqrt{\mathrm{count}(T)}}.$$

This UCB-style term grants larger bonus to under-tried tags and decays as a tag is reused.

**Information gain** To reward novelty relative to what has already been learned, we compare a candidate's node embedding to the centroid of accumulated learnings. Let $\mathbf{e}_\ell$ be the embedding of a learning $\ell$. For node $n$ with number of learnings $L(n) = \{\ell_i\}_{i=1}^{m(n)}$, define its embedding $\mathbf{e}_n = \frac{1}{m(n)} \sum_{i=1}^{m(n)} \mathbf{e}_{\ell_i}$ (when $m(n) > 0$). Let $\mathcal{L}$ be the set of all learnings gathered so far, $M = |\mathcal{L}|$, and

$\mu = \frac{1}{M} \sum_{\ell \in \mathcal{L}} \mathbf{e}_\ell$. Then

$$\mathrm{InfoGain}(n_k^\star) = \begin{cases} 1 - \mathrm{sim}\big(\mathbf{e}_{n_k^\star}, \mu\big), & m(n_k^\star) > 0 \text{ and } M > 0, \\ 0, & m(n_k^\star) = 0, \\ 1, & \text{otherwise.} \end{cases}$$

**Execution cost** Let $D$ be the max depth, $d(n)$ the depth of node $n$, and $K$ the branching factor. For child $n_k^\star$, the remaining depth is $d_{\mathrm{rem}} = D - d(n_k^\star)$. The number of nodes in a saturated $K$-ary subtree is

$$N_{\mathrm{rem}} = \begin{cases} \dfrac{K^{d_{\mathrm{rem}}+1} - 1}{K - 1}, & K > 1, \\ d_{\mathrm{rem}} + 1, & K = 1. \end{cases}$$

With a running average token cost $\mathrm{Tok}_{\mathrm{avg}}$ per node, the estimated tokens are $T_k^{\mathrm{est}} = \mathrm{Tok}_{\mathrm{avg}} N_{\mathrm{rem}}$, and the normalized execution cost is

$$C^{\mathrm{exec}}(n_k^\star) = \frac{T_k^{\mathrm{est}}}{T_k^{\mathrm{est}} + \mathrm{Tok}_{\mathrm{avg}}} = \frac{N_{\mathrm{rem}}}{N_{\mathrm{rem}} + 1}.$$

**Filtering candidates when pausing** Let $U_k = U(n_k^\star)$ and $conf_k \in [0, 1]$ be a confidence score generated by the LLM (see Appendix K, *Search Result Processing* prompt). Define the uncertainty radius

$$r_k = (1 - conf_k)\big(\max_{i \in K} U_i - \min_{i \in K} U_i\big),$$
$$U_k^{\mathrm{upper}} = U_k + r_k, \quad U_k^{\mathrm{lower}} = U_k - r_k.$$

The *could-be-the-best* set is

$$S = \big\{k \mid U_k^{\mathrm{upper}} \geq \max_{i \in K} U_i^{\mathrm{lower}}\big\}.$$

This mirrors upper and lower confidence bounds for best-arm filtering (Jamieson et al., 2014).

## E. Data Construction Details

To evaluate our method, we need a dataset with deep research worthy questions paired with realistic personas, where personas are, as defined in Section 2.1, $(p_{\mathrm{text}}, \mathcal{A}),$

where $p_{text}$ is a string, combining the user's profile and personality, and $\mathcal{A}$ is a set of *aspects* that the user is interested to see in a high-quality, well-aligned final report. We construct our dataset on top of the subset of 1,000 queries from Researchy Questions dataset (Rosset et al., 2024) used in DeepResearchGym (Coelho et al., 2025).

For each query, we first generate one or more ($p_{text}$ that would be reasonable to ask the query. For this, we adopt a two-step approach. In the first step, inspired by Wu et al. (2025) and (Wang et al., 2023), we use an iterative self-generation and filtering pipeline. In each round, 3 profiles are randomly selected from the profiles in the ALOE dataset (Wu et al., 2025) and used as input to an off-the-shelf LLM (GPT-4o) to generate 3 new profiles that would be reasonable to ask the query per iteration. Then we introduce an automatic filtering process based on semantic similarity to ensure the distinctiveness and diversity of the generated profiles. Same as Wu et al. (2025), we use Sentence Transformers (Reimers & Gurevych, 2019) to compute embedding of the generated profiles and measure the cosine similarity among the generated new profiles. For each new profile, if the highest similarity score compared to the other profiles exceeds 0.65, the profile is considered too similar to at least one of the other profiles and discarded. Otherwise, it will be accepted as a successful new profile to pair with the query. We repeat the process until 3 new accepted profiles are generated. In step 2, for each accepted profile, we generate a reasonable personality with GPT-4o to pair with it. For this, we randomly sample personalities from personality pool of the ALOE dataset as sample personalities fed into the LLM for generation.

Once we have generated one or more $p_{text}$ for each query, we then generate the set of aspects $\mathcal{A}$ for each $p_{text}$. We adopt the same approach as in Salemi & Zamani (2025) to generate 5-8 specific aspects that a user (described by $p_{text}$) would expect to see in a comprehensive and helpful report to the query, along with an evidence and a reasoning for each aspect, attributed from $p_{text}$.

Table 5 details the statistics of our generated dataset. All prompts for persona/profile/aspect generation are provided in Appendix L.

# F. Additional Experiment Details

## F.1. Additional Metrics

Table 6 reports sentence-level focus and DeepResearchGym relevance (support ↑ and contradiction ↓). We do not use sentence-level focus as a primary metric because it is length sensitive: the score $Focus_{st}$ is the fraction of sentences mapped to any aspect, so longer reports with a few connective or background sentences are penalized, whereas terse styles can inflate the ratio. Still, STEER achieves

competitive values (e.g., 80.67 at $C_0=0.1$), on par with the proprietary model and higher than the open-source baselines, indicating that personalization does not come at the cost of sentence-level topicality.

DeepResearchGym relevance compares a report to a pre-extracted, task-generic keypoint list; because STEER steers into personalized directions, it is expected to score lower on relevance$_{sup}$ than a system optimized for the generic keypoints (e.g., o4-mini-deep-research), while maintaining moderate relevance$_{con}$. In our results, STEER's relevance$_{sup}$ is similar to GPT-Researcher and OpenDeepResearch, with contradiction around 1.10–1.13; the proprietary model attains higher support but also substantially higher contradiction, whereas OpenDeepResearch shows low contradiction but lower support. Taken together, these metrics are complementary diagnostics: sentence-level focus confirms topicality at the sentence granularity, and DeepResearchGym relevance reflects overlap with generic keypoints rather than user-specific goals.

## F.2. Base Pause Cost vs. Pause Behavior

To better understand system behavior, Figure 2 shows the distribution of number of pauses per run. As expected, lowering base pause cost increases the number of pauses, with median pauses dropping from around 10 ($C_0 = 0.0$) to fewer than 2 ($C_0 \geq 0.8$). Compared to an LLM-based PauseAgent baseline, which issues many more questions, STEER's cost-sensitive mechanism achieves tighter control over the frequency of interruptions. This suggests that base pause cost provides a direct and interpretable knob for regulating user burden.

## F.3. STEER's Persona Modeling Analysis

To assess the effectiveness of STEER's dynamic persona modeling, we examine how well the inferred persona aligns with the system's final report over the course of interaction. Specifically, we track the alignment score between the generated report and the inferred persona at different base pause cost ($C_0$) settings, alongside the alignment between the report and the ground-truth persona provided at the start.

**Report Alignment Tracks Persona Alignment** As shown in the right panel of Figure 7, there is a strong positive correlation between STEER's report alignment and the alignment of its inferred persona to the ground-truth aspect set. The Pearson correlation is $r = 0.85$ ($p = 8.7 \times 10^{-4}$), indicating that improvements in inferred persona accuracy are tightly coupled with improvements in report quality. This supports the intuition that STEER's performance stems not only from architectural advances like mid-process pausing, but also from its ability to incrementally build an accurate model of user goals.

| System | Focus$_{st}$ ↑ | Relevance$_{sup}$ ↑ | Relevance$_{con}$ ↓ |
|---|---|---|---|
| GPT-Researcher | 67.07 | 61.39 | 1.02 |
| GPT-Researcher$_{initial-persona}$ | 69.18 | 60.82 | 1.11 |
| GPT-Researcher$_{full-persona}$ | 70.78 | 59.94 | 1.04 |
| OpenDeepResearch | 73.15 | 60.36 | **0.69** |
| OpenDeepResearch$_{initial-persona}$ | 75.61 | 57.07 | 0.81 |
| OpenDeepResearch$_{full-persona}$ | 78.84 | 57.06 | 0.81 |
| o4-mini-deep-research$_{initial-persona}$ | 78.41 | **67.36** | 1.74 |
| o4-mini-deep-research$_{full-persona}$ | 80.60 | 66.45 | 1.94 |
| **STEER**$_{[C_0=0.7]}$ | 78.51 | 60.47 | 1.13 |
| **STEER**$_{[C_0=0.1]}$ | **80.67** | 60.19 | 1.10 |

*Table 6.* Performance comparison between **STEER** and baseline frameworks on sentence-level focus score and report relevance scores.

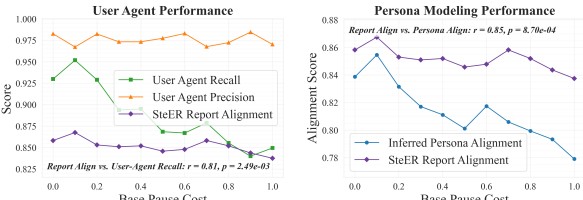

*Figure 7.* Analysis of User Agent and Persona Modeling Performance across Base Pause Cost ($C_0$). *Left:* User Agent precision, recall, and **STEER** report alignment scores plotted across varying base pause cost values. *Right:* Alignment scores of **STEER**'s inferred persona and final report, both evaluated against the ground-truth aspect set $\mathcal{A}$, plotted across varying base pause cost values.

**Impact of Base Pause Cost** We observe a general downward trend in both inferred persona alignment and report alignment as base pause cost increases. This confirms that higher interruption costs reduce the frequency of clarifying interactions, resulting in less accurate persona estimates and, consequently, less aligned outputs. In contrast, low $C_0$ values allow **STEER** to query the user more frequently, leading to refined persona inference and stronger downstream alignment.

These results highlight the central role of interactive refinement in personalized research workflows. Rather than relying solely on upfront persona injection, **STEER** learns about the user incrementally — and this process is empirically shown to improve alignment. The correlation between inferred and actual persona alignment validates the design of our live persona model and its integration into the decision-making process.

### F.4. User Agent Performance Analysis

To enable scalable, automatic evaluation of **STEER**, we employ a User Agent that simulates a real user interacting with the system. This User Agent is responsible for selecting preferred research directions based on a target persona and proposing new follow-up questions when relevant aspects

remain uncovered. Its effectiveness directly impacts the utility of our offline evaluation framework.

As shown in the left panel of Figure 7, the User Agent maintains consistently high precision across a wide range of $C_0$ values, with scores above 0.97. This suggests that when the agent chooses to retain a direction, it is highly likely to align with the user's intended aspects. In contrast, recall is more sensitive to the pausing configuration. At lower $C_0$ (e.g., 0.1), the User Agent achieves peak recall near 0.95, but recall steadily declines as $C_0$ increases, falling to approximately 0.85 by $C_0 = 1.0$. This reflects the agent's conservative behavior under higher interruption costs, where it refrains from selecting additional directions that could be beneficial.

We also observe that the alignment score of the final report generated by **STEER** (in purple) closely tracks the recall curve of the User Agent. The Pearson correlation between the two is strong and statistically significant ($r = 0.81$, $p = 2.49 \times 10^{-3}$), as annotated in the plot. This indicates that the breadth of information the agent retains during interaction is highly predictive of the alignment quality of the final report. The stronger the agent's coverage of relevant aspects (recall), the more aligned the report tends to be with the user's needs.

These results confirm that the simulated User Agent is not only a faithful proxy for real user behavior but also a critical driver of **STEER**'s alignment performance. Its high precision ensures quality, while its recall effectively governs how much of the user's goals are ultimately realized in the research output.

## G. User Study Details

To complement automated evaluation, we conducted a human annotation study to directly assess how well **STEER** reports align with user personas compared to baseline systems. We developed a custom web-based annotation platform (Figure 8) that guides annotators through a structured

evaluation procedure with clear instructions and embedded report viewers.

## G.1. Setup

Annotators were provided with a **persona card** containing (i) the query, (ii) a short persona description, and (iii) the persona's **interested aspects**—the specific information needs that the final report should cover. These interested aspects formed the primary basis of evaluation. Annotators then evaluated two reports for the same query—persona pair: one generated by **STEER** and one by a baseline system (either GPT-Researcher or Open Deep Research). Report order was randomized to reduce bias.

## G.2. Evaluation Procedure

- **Step 1: Aspect Coverage.** Annotators skimmed both reports and rated, for each aspect, how well the report addressed it on a 3-point scale: **0 = not covered**, **1 = somewhat covered**, **2 = fully covered**. When assigning a score of 1 or 2, annotators were instructed to copy-paste a short supporting quote (1—2 sentences) from the report to ground their judgment. This ensured ratings were evidence-backed rather than impressionistic.

- **Step 2: Findability.** Annotators rated how easy it was to locate content relevant to each aspect in the report on a 3-point scale: **0 = difficult**, **1 = medium**, **2 = easy**. This step captured not only whether the aspect was present, but also whether it was readily discoverable by a reader.

- **Step 3: Report Comparison.** Based on their coverage and findability assessments, annotators selected a winner between the two reports along two dimensions: **Alignment** (which report better served the persona's aspects) and **Focus** (which report stayed more on-topic versus digressing into irrelevant content).

**Interface Design.**  The interface (Figure 8) displayed both reports side by side in embedded PDF viewers, alongside the persona's aspects in a draggable panel for quick reference. Each evaluation step was clearly separated into dedicated panels, with concise instructions and tips (e.g., "**You don't need to read every word**—scan section titles and opening sentences for relevant content"). Progress indicators guided annotators through the sequence, ensuring consistency. Importantly, the platform emphasized that judgments should be made **from the persona's perspective**, not based on annotators' personal preferences.

**Instructions and Quality Control.**  The study followed a three-step protocol:

As displayed in Figure 9, annotators were instructed to:

1. Read the persona aspects carefully, treating them as the ground truth for evaluation.

2. Provide evidence quotes for all non-zero aspect coverage ratings.

3. Complete all steps in sequence (coverage $\rightarrow$ findability $\rightarrow$ comparison).

4. Judge strictly by persona relevance, not by report verbosity, formatting, or personal opinion.

These safeguards helped ensure high-quality, reproducible annotations grounded in persona-aligned judgments.

**Blinding considerations.**  Pairwise comparison in open-ended generation cannot guarantee perfect blinding, and we treat this as a known limitation rather than a solved problem. We took several steps to reduce identifiability risk: (i) annotators were shown only the two final reports, side by side in randomized order, with no access to clarification questions, user-system interaction traces, persona updates, or any other intermediate system behavior; (ii) all systems produced report-style outputs with broadly similar structure — e.g., tables of contents and section headings appeared across both **STEER** and the baselines — so we did not observe consistent structural giveaways; and (iii) annotators were unfamiliar with our system or the baselines' formatting conventions and were instructed to judge from the persona's perspective rather than from stylistic preference. Residual risk remains: a sufficiently distinctive lexical or structural fingerprint could in principle be guessed, and we cannot rule out that some annotators occasionally formed such a guess. We therefore interpret the human-study results as strong but not unconfounded evidence, and view fully blinded protocols (e.g., post-hoc style normalization or third-party rewriting) as useful future work.

## H. STEER Working Prototype

To illustrate the functionality of **STEER**, we build an interactive web-based prototype (Figure 10) that visualizes the **STEER** framework in action. The interface consists of three synchronized panels: **(i)** a conversation pane for clarification prompts and user feedback, **(ii)** a dynamically expanding research tree that reflects the research status and partial research results, and **(iii)** a live persona tracker that displays the evolving inferred persona $\hat{P}$ and monitors the updating alignment between cumulative research results and the inferred user aspects $\hat{A}$. This prototype supports interactive research sessions, allowing users to guide the exploration by selecting preferred subtopics or introducing new follow-up questions mid-process.

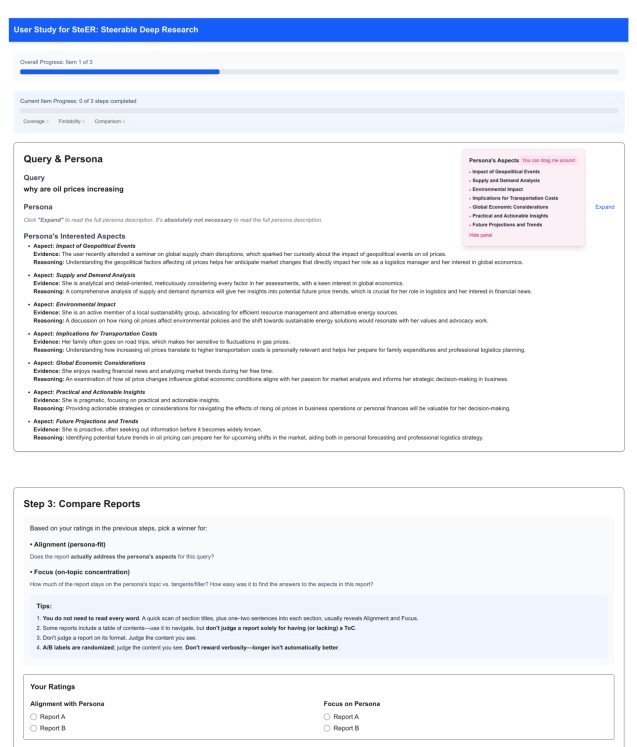
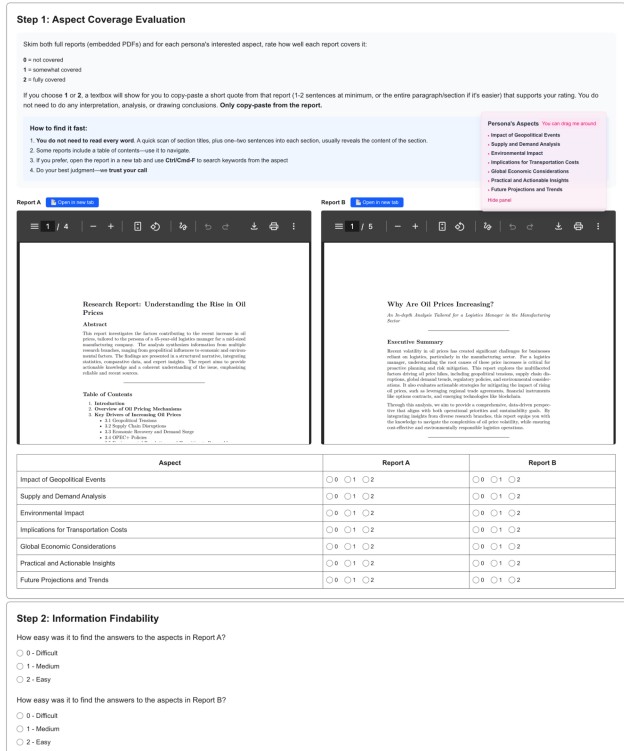

*Figure 8.* User study interface.

# I. LLM-as-Judge Evaluation

To validate the effectiveness of the LLM judge used throughout evaluation, we conduct a small-scale meta-evaluation of the LLM-as-judge. Specifically, we take the alignment score per aspect produced by the LLM judge (*gpt-4.1-mini*) and the Coverage score per aspect produced by human annotators in the user study (both in the scale of 0 - 2), and compute the Pearson correlation $r$ between LLM-assigned and human-assigned scores over all overlapping aspects where both annotations are available. We obtain $r = 0.34$, $p$-value $< 0.0001$, indicating a statistically significant, moderate positive correlation between the LLM-as-judge and human evaluations. At the annotator level, all annotators exhibit positive correlations with the LLM judge, ranging from 0.19 to 0.45 with small variability (standard deviation = 0.094), suggesting that the *gpt-4.1-mini*'s scoring is consistently aligned with different human raters rather than being driven by any single annotator. While imperfect, these results indicate that the LLM-judge is directionally consistent with human judgments at the aspect level and is suitable as a scalable proxy for our large-scale evaluation, especially when interpreted alongside the human study that directly validates our main claims.

# J. Limitations and Future Work

**Simulated personas and queries vs. users' own queries.** Our user study asks annotators to *mimic* a target persona and query rather than using their own information needs. This is intentional: our goal is to evaluate system-level behavior under controlled, persona-conditioned information needs and to compare STEER against baselines on *exactly* the same persona–query pairs. This requires *(i)* a fixed set of personas and queries that all systems answer and *(ii)* a shared reference for annotators, so that cross-system differences in alignment and coverage can be attributed to the system rather than to heterogeneous user goals. If each participant were to choose their own query and implicit persona, different systems would be evaluated on tasks of varying difficulty, domain, and specificity, making it difficult to perform clean system-level comparisons and to interpret differences in outcomes as stemming from the model rather than from the task. Moreover, obtaining enough repeated measurements per system–persona–task condition under fully free-form user queries would require a substantially larger number of participants and interactions, constituting a much larger user-study effort in terms of human resources, annotation time, and cost. We acknowledge that this controlled design does not fully capture all aspects of real-world usage, particularly long-term adaptation to an individual's genuine information needs. As outlined in our limitations and future

work, a natural next step is to conduct more comprehensive, end-to-end user studies in which participants bring their own personas and tasks, interact with the system over longer sessions, and are evaluated on richer metrics such as task success, time to insight, perceived control and trust, and cognitive load.

**Blinding in pairwise human evaluation.** Our user study compares final reports from **STEER** and a baseline side by side in randomized order (Appendix G). Annotators see only the final reports — never clarification turns, interaction traces, or intermediate system state — and we did not observe consistent structural cues separating systems. However, perfect blinding is difficult in any open-ended generation setting: lexical, formatting, or organizational tendencies of the underlying model and pipeline could in principle leak system identity. We therefore view the pairwise preferences as supportive but not fully unconfounded evidence, complementing the controlled, persona-conditioned automatic evaluation. Stronger blinding (e.g., style-normalized rewrites, third-party paraphrasing, or A/B/X protocols) is a natural next step.

**Robustness to noisy and contradictory user feedback.** Our User Agent is, by construction, a high-fidelity simulator: it stays consistent with a fixed target persona, responds to every clarification, and does not fatigue. Real users behave differently — they may give contradictory or self-revising answers, supply partial or vague responses, ignore some prompts, or abandon the interaction entirely if asked too much. **STEER**'s live persona module already handles *new* preferences by updating the inferred persona from incoming feedback, but it does not explicitly resolve *conflicts* between earlier and later signals, nor does it model the cost of disengagement when interactions accumulate. We see three concrete extensions as important future work: (i) preference-conflict handling that reconciles contradictory feedback over time, drawing on agent memory and preference-tracking literature; (ii) graceful degradation under partial or missing responses, so the system can still steer with low-confidence signals rather than treating silence as endorsement; and (iii) abandonment-aware pause policies that incorporate user fatigue or drop-out risk into the cost-benefit rule. Importantly, we view these as natural extensions of **STEER**'s interactive paradigm rather than departures from it — the same framework that opens the door to mid-process steering also opens the door to studying how users actually steer.

**Runtime and latency analysis.** A key limitation of this work is that we do not provide a systematic runtime or latency analysis of **STEER** relative to baseline frameworks. Because **STEER** is explicitly designed as an interactive deep-research system, defining and comparing "runtime" in a meaningful way is non-trivial: a naive wall-clock mea-

sure would conflate *(i)* time spent on user interaction (or User Agent responses in our simulations) and *(ii)* the highly variable size of the research tree induced by different pause policies and user choices. Simply averaging end-to-end runtimes could therefore be misleading — **STEER** may spend more time within a single run while reducing wasted effort by correcting misalignment earlier, whereas non-steerable baselines may require stacking multiple full deep-research calls to reach a comparable result. Moreover, even in a controlled environment like DeepResearchGym, end-to-end latency is heavily influenced by external factors such as search endpoint stability, network conditions, and rate limits, especially in multi-agent, tool-using pipelines. As a result, raw wall-clock differences are hard to attribute cleanly to the pause policy rather than infrastructure noise. We view a careful efficiency study — using user-centric measures such as time- or turns-to-satisfactory-answer, and stratifying by tree size, number of pauses, and backend conditions — as an important direction for future work, and plan to complement our current alignment and quality evaluations with such analyses in follow-up studies.

## K. Prompt Templates in STEER

We include here the core prompt templates used in our **STEER** framework, organized by functionality. Each block shows the **system prompt** and the corresponding **user prompt**. Placeholders such as {query}, {persona_text}, and {checklist_items} are substituted at runtime.

**Research Planning**

```
System Prompt
You are an expert researcher working with a
    specific user persona. Your task is to analyze
    the original query and search results, then
    generate targeted questions that explore
    different directions and time periods of the
    topic, specifically tailored to the user's
    interests and checklist items.

User Prompt
Original query: \{query\}

Current time: \{current_time\}

User persona: \{persona_text\}

User checklist (aspects they care about):
\{checklist_items\}

Search results:
\{search_results\}

Based on these results, the original query, the
    user's persona, and their checklist, generate
    5-8 unique follow-up questions that:
1. Explore different directions relevant to this
    query
2. Cover a good wide range of topics and aspects of
    the query
3. Consider recent developments up to
    \{current_time\}
```

```
4. Are somewhat tailored to the user's background
      and needs, but not constrained by the user's
      persona and interests
5. Each follow-up question should cover a distinct
      thematic facet – do not repeat other questions

For each question, provide a confidence score
      between 0.0 and 1.0 indicating:
- Relevance of the question to the main research
      query
- Insightfulness of the question that would be
      useful for the final report generation
- How likely this question is to lead to valuable
      information for this user

Return your response as a JSON object with the
      following structure:
\{
  "follow_up_questions": [
    \{
      "follow_up_question": "follow-up question
    text",
      "confidence": 0.0-1.0,
      "reasoning": "why this is a good follow-up
    question"
    \}
  ]
\}
```

## Search Result Processing

**System Prompt**
```
You are an expert researcher analyzing search
      results for a specific user persona. Focus on
      extracting learnings and follow-up questions
      that are most relevant to the user's interests
      and checklist items.
```

**User Prompt**
```
Given the following research results for the query
      '\{query\}', extract key learnings and suggest
      5-8 follow-up questions that are specifically
      relevant to the user's persona and interests.

User persona: \{persona_text\}

User checklist (aspects they care about):
\{checklist_items\}

Previously seen tags: \{seen_tags\}

Focus on:
1. Learnings that address the user's checklist items
2. Information relevant to their background and
      interests
3. Follow-up questions that would help address
      their specific needs
4. Each follow-up question should cover a distinct
      thematic facet – do not repeat other questions

For each follow-up question, provide a confidence
      score between 0.0 and 1.0 indicating:
- How likely this question is to lead to valuable
      information for this user
- Alignment with user's persona and checklist items
- Relevance to the original research query

Additionally, ALWAYS generate one "wild-card"
      question in the separate wild_card_question
      field that goes outside the inferred persona
      but is plausibly useful for broader
      understanding of the topic.

Additionally, assign tags to categorize what
      aspects this research content covers. Tags
      should be short phrases (2-4 words) that
      describe the key topics, themes, or domains
      covered by the query, context, and learnings.
```

```
      Be very cautious about adding new tags:
- FIRST, check if any of the previously seen tags
      are relevant to this content
- REUSE existing tags whenever possible
- ONLY add new unseen tags if the content covers
      aspects not captured by existing tags
- Keep tags concise and descriptive

Return your response as a JSON object with the
      following structure:
\{
  "learnings": [
    \{
      "insight": "key insight or finding relevant
    to the user",
      "source_url": "URL of the source (if
    available)",
      "relevance_to_user": "how this learning
    relates to the user's interests"
    \}
  ],
  "follow_up_questions": [
    \{
      "follow_up_question": "follow-up question
    text",
      "confidence": 0.0-1.0,
      "reasoning": "why this is a good follow-up
    question"
    \}
  ],
  "wild_card_question": \{
    "question": "wild-card question that goes
    outside the persona but is plausibly useful",
    "confidence": 0.0-1.0,
    "reasoning": "why this wild-card question could
    be valuable"
  \},
  "tags": ["tag1", "tag2", "tag3"]
\}

Research query: \{query\}

Search results:
\{context\}
```

## Follow-up Questions to Search Queries

**System Prompt**
```
You are an expert search query optimizer. Your task
    is to convert follow-up research questions
    into effective search queries that will yield
    relevant search results.
```

**User Prompt**
```
Convert the following follow-up question into an
      optimized search query that will yield
      relevant search results.

Original research query: \{original_query\}

User persona: \{persona_text\}

User checklist (aspects they care about):
\{checklist_items\}

For each of the following follow-up question,
      create a search query that:
1. Effectively searches for information to answer
      the follow-up question
2. Is optimized for search engines
3. Maintains connection to the original research
      query
4. Considers the user's persona and interests

For each search query, also provide a clear
      research goal that describes:
- What specific information or insights this search
      aims to discover
```

```
- How it relates to the original research question
- What direction of the topic it will explore

Follow-up questions:
\{followup_questions\}

Return your response as a JSON object with the
    following structure:
\{
  "search_queries": [
    \{
      "follow_up_question": "input follow-up
    question text",
      "search_query": "optimized search query",
      "research_goal": "clear description of what
    this search aims to discover and how it
    relates to the original research question"
    \}
  ]
\}
```

## Persona Checklist Inference

**System Prompt**
You are an expert at understanding user personas
    and inferring what aspects they would care
    about in research responses. Your task is to
    analyze a user's persona and generate specific
    checklist items they would expect to see
    addressed.

**User Prompt**
Given the following user persona and their research
    query, infer a checklist of specific aspects
    that this user would expect to see addressed
    in a comprehensive research response.

User persona: \{persona_text\}

Research query: \{query\}

Based on this persona and query, generate 5-8
    specific checklist items that this user would
    expect to see in a helpful response. Each item
    should be:
1. Specific to this user's background and interests
2. Relevant to the research query
3. Actionable and measurable
4. Distinct from other items

Return your response as a JSON object with the
    following structure:
\{
  "checklist_items": [
    "specific aspect this user would expect to see
    addressed",
    "another specific aspect relevant to their
    interests"
  ]
\}

## Persona Modeling

**System Prompt**
You are an expert at understanding user personas
    and updating them based on user interactions.
    Your task is to analyze a user's response and
    infer additional information about their
    persona and interests.

**User Prompt**
Given the following current persona and a user's
    response to a research proposal, infer
    additional information about this user's
    persona and interests.

Current persona: \{current_persona\}

Current checklist items they already care about:
\{current_checklist\}

User's response: \{user_response\}

Based on this response, identify additional
    information about the user's:
1. Background and interests
2. Specific preferences and priorities
3. Communication style and concerns
4. Any new aspects they care about

IMPORTANT: Do NOT output repetitive information:
- Only include NEW persona information that isn't
    already covered in the current persona
- Only include NEW checklist items that aren't
    already in the current checklist
- If nothing new can be inferred, return empty
    strings and empty arrays

Return your response as a JSON object with the
    following structure:
\{
  "additional_persona_info": "new information to
    append to the persona (empty if nothing new)",
  "new_checklist_items": [
    "new aspect they mentioned or implied they care
    about (only if not already in checklist)",
    "another new aspect if applicable"
  ]
\}

## Clarification Question Generation

**System Prompt**
You are an expert research assistant. Your task is
    to generate clear, helpful clarification
    questions that present concise summaries of
    research directions to users for selection.

**User Prompt**
Based on the following research context, generate a
    structured clarification question that
    presents concise summaries of the available
    research directions to the user for selection.

Research query: \{query\}

Available research directions (search queries):
\{research_directions\}

User persona: \{persona_text\}

Create a structured question that:
1. Starts with a natural introduction
2. Lists each research direction as numbered bullet
    points (1., 2., 3., etc.)
3. For each direction, provide a concise summary (1
    sentence) that captures the essence of what
    that search query would explore, rather than
    showing the raw search query
4. Provides clear selection instructions:
   - To select directions: just type the bullet
    numbers (e.g., "1, 3")
   - To suggest new follow-up questions: start a
    new line with "New follow-up questions:"
    followed by each new follow-up question on
    separate lines
5. Matches the user's communication style

Return your response as a JSON object with the
    following structure:
\{
  "clarification_question": "your structured
    question to the user with concise summaries"
\}
```

## Report Generation

**System Prompt**
You are a professional research report writer specializing in persona-aware reports. Create comprehensive, well-structured reports based on research data with proper citations, tailored to the specific user's background and interests.

**User Prompt**
Using the following hierarchically researched information and citations:

"\{context\}"

Write a comprehensive research report answering the query: "{question}"

User persona: \{persona_text\}

User interests (checklist items they care about):
\{checklist_items\}

The report should:
1. Synthesize information from multiple levels of research depth
2. Integrate findings from various research branches
3. Present a coherent narrative that builds from foundational to advanced insights
4. Maintain proper citation of sources throughout
5. Be well-structured with clear sections and subsections
6. Have a minimum length of \{total_words\} words
7. Follow \{report_format\} format with markdown syntax
8. Use markdown tables, lists and other formatting features when presenting comparative data, statistics, or structured information
9. Be tailored to the user's persona and interests

Additional requirements:
- Prioritize insights that emerged from deeper levels of research
- Highlight connections between different research branches
- Include relevant statistics, data, and concrete examples
- Focus on directions that align with the user's interests and checklist
- Use language and explanations appropriate for the user's background
- Address the user's specific concerns and priorities
- You MUST determine your own concrete and valid opinion based on the given information. Do NOT defer to general and meaningless conclusions.
- You MUST prioritize the relevance, reliability, and significance of the sources you use. Choose trusted sources over less reliable ones.
- You must also prioritize new articles over older articles if the source can be trusted.
- Use in-text citation references in \{report_format\} format and make it with markdown hyperlink placed at the end of the sentence or paragraph that references them like this: ([in-text citation](url)).
- Write in \{language\}

Citation requirements:
- You MUST write all used source URLs at the end of the report as references
- Make sure to not add duplicated sources, but only one reference for each
- Every URL should be hyperlinked: [url website](url)
- Include hyperlinks to the relevant URLs wherever they are referenced in the report
- Format example: Author, A. A. (Year, Month Date). Title of web page. Website Name. [url website](url)

Please write a thorough, well-researched report that synthesizes all the gathered information into a cohesive whole, tailored specifically to this user's persona and interests.
Assume the current date is \{current_date\}.

## Persona Alignment Evaluation

**System Prompt**
You are an expert evaluator specializing in assessing how well research content aligns with user personas and interests. Your task is to analyze content and determine how well it addresses specific directions important to the user.

**User Prompt**
You are evaluating how well research content aligns with a user's persona and interests.

# User Persona: \{persona_text\}

# Research Content:
\{content\}

# Key Learnings:
\{learnings\}

# Checklist Items to Evaluate:
\{checklist_items\}

For each checklist item, evaluate how well the research content and learnings address it.
Provide a score from 0-2 for each item:
- 0: Not addressed or covered
- 1: Somewhat addressed or partially covered
- 2: Well addressed or thoroughly covered

Return your evaluation as a JSON object with the following structure:
```
\{
  "evaluations": [
    \{
      "item": "checklist item text",
      "score": 0-2,
      "reasoning": "brief explanation of the score"
    \}
  ]
\}
```

# L. Prompt Templates for Data Generation

We include here the prompt templates used in data generation.

## Profile Generation Prompt

**User Prompt**
Generate a user profile for someone who would logically and reasonably ask the following question: "\{query\}"

The profile should include demographic and background information such as age range, occupation, hobbies, family structure, education background, or any other relevant facts. Note that you don't need to include all of these details for each persona. You can use any kinds of combinations and please think about other aspects other than these.You should include something that can be elicited from daily and natural conversations. You should not include too much information about

```
       this person's work content and you should not
       give any description about the user's
       personality traits. Focus on objective facts
       about the person.

   Here are some example profiles for reference:
   \{profile_examples\}

   Generate a single user profile that contains around
       8-10 distinct facts about the person. The
       profile should logically connect to why this
       person would ask the given question. You
       should only output the profile in plain text
       format.

   IMPORTANT: Try to be creative and comprehensive.
       Make sure the profile makes it realistic for
       this person to ask the specific question.
```

### Personality Generation Prompt

```
User Prompt
Generate personality traits for a person with the
    following profile who would ask this question:
    "\{query\}"

Profile:
\{generated_profile\}

Based on this profile and the question they would
    ask, generate appropriate personality traits.
    You should include something that can be
    elicited from daily and natural conversations.
    Each description should contain around 8-10
    personality traits about the person.

Here are some example personality descriptions for
    reference:
\{personality_examples\}

Generate personality traits that are consistent
    with the profile and make it logical for this
    person to ask the given question. You should
    only output the personality description in
    plain text format.

IMPORTANT: You should not include any other content
    that is beyond personality traits, such as
    occupation or demographic information (those
    are already in the profile). Focus only on
    personality characteristics, behavioral
    patterns, and psychological traits. Be
    creative and make sure the personality aligns
    with both the profile and the research
    question.
```

### Aspect Generation Prompt

```
User Prompt
Given a user's persona and their query, generate a
    list of specific aspects that this user would
    expect to see addressed in a high-quality
    response to their query. These aspects will
    serve as evaluation criteria to assess how
    well a response meets this specific user's
    needs and expectations.

Query: "\{query\}"

User Persona:
\{persona\}

Based on this persona and query, generate 5-8
    specific aspects that this user would expect
    to see in a comprehensive and helpful
    response. Each aspect should be:
```

```
1. Specific to this user's background, needs, and
      context
2. Actionable and measurable (can be used to
      evaluate a response)
3. Relevant to the query and persona
4. Distinct from other aspects (no overlap)

Format your response in JSON format where each
      aspect is a clear, specific expectation that
      can be used to evaluate whether a response
      adequately addresses this user's needs and
      provide a clear explanation of why each aspect
      is significant for the user and what specific
      details they would expect to see in the
      response. Focus on what content, depth, style,
      or approach would be most valuable for this
      specific user.

Each aspect should have the following fields:
- aspect: a string that is the name of the aspect
      that is important to be present in the response
- evidence: a string that points to specific
      details from the user's persona that indicate
      this aspect is important
- reason: a string that explains why the aspect is
      important for the user

Use the following JSON structure:
\{
  "aspects": [
    \{
      "aspect": "Name of the aspect",
      "evidence": "Specific details from the user's
      persona that indicate this aspect is
      important",
      "reason": "Explanation of why this aspect is
      important for the user"
    \}
  ]
\}

IMPORTANT: Make the aspects specific to this user's
      unique situation and needs, not generic
      aspects that would apply to any user asking
      this question.
```

## M. Prompt Templates for Evaluation

We include here the prompt templates used in evaluation
scripts.

### Alignment Evaluation Prompt

```
User Prompt
You are a fair and insightful judge with
    exceptional reasoning and analytical
    abilities. Your task is to evaluate a user's
    question, a generated response to that
    question, and multiple aspects that are
    important to the user. Based on this
    information, assess how well each aspect is
    addressed in the generated response. Provide a
    clear and accurate assessment for each aspect.

# Your input:
- question: the question asked by the user
- persona: the user's persona (profile and
    personality) that the aspects are based on
- response: a generated response to the user's
    question
- aspects: a list of aspects that are important to
    the user, each consisting of:
  - aspect: the title for the aspect
  - reason: the reason that this aspect is
    important for the user
```

```
   - evidence: the evidence from the user persona
     that the aspect was extracted from

# Your output:
Your output should be a valid JSON object in
     ```json ``` format containing the following
     fields:
- evaluations: A list of evaluations for each
     aspect, where each evaluation contains:
  - aspect: the aspect title
  - match_score: A score between 0 to 2 that
    indicates how well the generated response
    addresses this aspect, where:
    * 0 means the response does not cover this
      aspect
    * 1 means the response somewhat covers this
      aspect
    * 2 means the response covers this aspect very
      well
  - reasoning: A detailed explanation of why this
    score was assigned, including specific
    examples from the response

# Question: \{question\}

# Persona: \{persona\}

# Response: \{response\}

# Aspects:
\{aspects_formatted\}

Output:
```

## Sentence Focus Evaluation Prompt

```
User Prompt
You are an expert judge evaluating whether
     sentences in a report cover specific user
     aspects. For each sentence, determine which
     aspects (if any) it addresses.

# Your input:
- sentences: numbered sentences from a report
- aspects: user aspects with IDs, titles, and
     reasons

# Your task:
For each sentence, identify whether it covers any
     of the user aspects. **BE EXTREMELY STRICT**
     in your evaluation.

A sentence covers an aspect ONLY if it:
1. Directly addresses the specific concern or
     interest described in the aspect
2. Provides substantive, detailed information that
     would be valuable to someone with that
     specific aspect
3. Goes beyond mere keyword mentions or general
     background information

A sentence does NOT cover an aspect if it:
- Only provides general background or introductory
     information
- Mentions keywords related to the topic but
     doesn't address the specific concern
- Gives broad overviews without targeting the
     particular interest
- Describes general principles without connecting
     to the specific aspect
- Is just factual information that doesn't serve
     the user's particular need

**Default to NOT covering aspects unless there is
     clear, direct, substantial relevance to the
     specific user concern.**

# JSON Schema for output:
```

```
\{
  "type": "object",
  "patternProperties": \{
    "^\d+\$": \{
      "type": "object",
      "properties": \{
        "cover_aspects": \{
          "description": "A list of aspect IDs that
    the sentence covers. If the sentence does not
    cover any of the aspects, the list should be
    empty.",
          "type": "array",
          "items": \{"type": "integer"\}
        \},
        "reasoning": \{"type": "string"\}
      \},
      "required": ["cover_aspects", "reasoning"]
    \}
  \}
\}

# Sentences:
\{sentences_formatted\}

# Aspects:
\{aspects_formatted\}

Output valid JSON:
```

## Key Point Extract Prompt

```
User Prompt
Based on the report provided, identify key points
     in the report that directly help in responding
     to the query. The key points are not simply
     some key content of the text, but rather the
     key points that are important for **answering
     the query**. IMPORTANT: Ensure each point is
     helpful in responding to the query. Keep the
     point using the original language and do not
     add explanations. IMPORTANT: Each span must be
     a single consecutive verbatim span from the
     corresponding passages. Copy verbatim the
     spans, don't modify any word! Your response
     should state the point number, followed by its
     content, and spans in the text that entail the
     key point. Respond strictly in JSON format:
\{
  "points": [ \{
    "point_content": point_content,
    "spans": [span1, span2, ...]
  \}, ... ]
\}

Remember:
- Key points can be abstracted or summarized, but
     the span must be a copy of the original text.
     The content of the key point does NOT need to
     be the same as that of the span.
- These keypoints must be helpful in responding
     tothe query.
- If thereare multiple spans for a point, add all
     of them in the spans list.

Report: \{report\}

Query: \{query\}

Output:
```

## Key Point Focus Evaluation Prompt

```
User Prompt
You are an expert judge evaluating whether key
     points of a report cover specific user aspects
     to answer a query. For each key point,
     determine which aspects (if any) it addresses.
```

```
     **BE EXTREMELY STRICT** in your evaluation.

A key point covers an aspect ONLY if it:
1. Directly addresses the specific concern or
    interest described in the aspect
2. Provides substantive, detailed information that
    would be valuable to someone with that
    specific aspect
3. Goes beyond mere keyword mentions or general
    background information

A key point does NOT cover an aspect if it only
    provides introductory information or broad
    overviews
**Default to NOT covering aspects unless there is
    clear, direct, substantial relevance to the
    specific user concern.**

Response strictly in JSON format:
\{
  "point_number": \{
    "cover_aspects": [aspect1, aspect2, ...],
    "reasoning": reasoning
  \},
\}

# Query:
\{query\}

# Report Key Points:
\{key_points_formatted\}

# UserAspects:
\{aspects_formatted\}

Output:
```

## User Agent Alignment Evaluation Prompt

```
User Prompt
You are a fair and insightful judge with
    exceptional reasoning and analytical
    abilities. Your task is to evaluate a user's
    follow-up questions in regard to a query, and
    multiple aspects that are important to the
    user. Based on this information, assess how
    well the follow-up questions trying to cover
    the user's interested aspects. An aspect is
    considered covered if there are follow-up
    questions are trying to initiate research
    directions that are related to the aspect.
    Provide a clear and accurate assessment for
    each aspect.

# Your input:
- query: the query asked by the user
- persona: the user's persona (profile and
    personality) that the aspects are based on
- follow-up questions: a list of follow-up
    questions that the user asked
- aspects: a list of aspects that are important to
    the user, each consisting of:
  - aspect: the title for the aspect
  - reason: the reason that this aspect is
    important for the user
  - evidence: the evidence from the user persona
    that the aspect was extracted from

# Your output:
Your output should strictly be a valid JSON object:
\{
  "evaluations": [ \{
    "aspect": aspect,
    "match_score": match_score,
    "reasoning": A detailed explanation of why this
     score was assigned, including specific
     examples from the follow-up questions
  \}, ... ]
```

```
\}

"match_score" is a score between 0 to 2 that
    indicates how well the follow-up questions
    addresses this aspect, where:
    * 0 means the follow-up questions does not
    cover this aspect
    * 1 means the follow-up questions somewhat
    covers this aspect
    * 2 means the follow-up questions covers this
    aspect very well

# Query: \{query\}

# Persona: \{persona\}

# Follow-up Questions:
\{follow_up_questions_formatted\}

# Aspects:
\{aspects_formatted\}

Output:
```

## User Response Precision Evaluation Prompt

```
User Prompt
You are an expert judge evaluating whether a user's
    follow-up questions or responses are truly
    targeted to specific user aspects for
    answering a query. For each follow-up,
    determine which aspects (if any) it
    substantively targets. BE EXTREMELY STRICT.

A follow-up COVERS an aspect ONLY if it:
1) Clearly aims to gather information directly
    relevant to the specific concern described by
    the aspect; AND
2) Goes beyond surface keywords or generic
    curiosity.

A follow-up does NOT cover an aspect if it:
- Is a broad/background question without tailoring
    to that aspect; OR
- Only mentions related keywords but lacks a
    targeted objective tied to the aspect; OR
- Is unrelated to the user's stated concerns.

Respond strictly in JSON format:
\{
  "response_number": \{
    "cover_aspects": [aspect_id_1, aspect_id_2,
    ...],
    "reasoning": reasoning
  \},
  ...
\}

# Query:
\{query\}

# User Responses (indexed from 0):
\{user_responses_formatted\}

# User Aspects (IDs start at 0):
\{aspects_formatted\}

Output:
```

## Final Persona State Evaluation Prompt

```
User Prompt
You are a fair and insightful judge with
    exceptional reasoning and analytical
    abilities. Your task is to evaluate how well
    items from a final persona state checklist
    cover user aspects. Given the user's query,
```

```
       the original persona, a list of checklist
       items, and the user aspects, assess for each
       aspect how well the checklist covers it.

# Your input:
- query: the query asked by the user
- persona: the user's original persona text
- checklist: a list of items inferred that might be
       important for the user to answer the query
- aspects: a list of aspects that are indeed
       important to the user as ground truth, each
       consisting of aspect, reason, and evidence

# Your output:
Return strictly valid JSON of the form:
\{
  "evaluations": [\{
    "aspect": aspect_title,
    "match_score": 0|1|2,
    "reasoning":
     detailed_reasoning_referencing_specific
     _checklist_items
  \}, ... ]
\}

Interpret match_score as:
- 0: the checklist does not cover this aspect
- 1: the checklist somewhat covers this aspect
- 2: the checklist covers this aspect very well

# Query: \{query\}

# Persona: \{persona\}

# Checklist Items:
\{checklist_formatted\}

# Aspects:
\{aspects_formatted\}

Output:
```

# N. Additional Prompt Templates

We include here the prompt templates used for User Agent
and Pause Agent.

## User Agent

```
System Prompt
You are simulating a real user with a specific
     persona and interests. Your task is to respond
     to SteER's research proposals by selecting
     relevant directions and suggesting new
     directions based on your persona and research
     interests.

User Prompt
You are acting as a user with the following persona:

# User Persona:
\{persona_text\}

# Aspects and directions You Care About:
\{aspects_text\}

# History of your previous asked questions:
\{questions_history_text\}

# Research Query:
\{query\}

# SteER's Proposal:
\{steer_proposal\}
```

```
SteER is presenting research directions as numbered
     bullet points. Based on your persona and
     interests, respond as this user would by:
1. Selecting ONLY the most relevant direction
     numbers that have the highest priority for
     this research
2. Suggesting new follow-up questions ONLY if you
     feel there's a very important direction
     missing from the proposal
3. Providing natural commentary as this user would
     speak

**IMPORTANT CONSTRAINTS:**
- **DO NOT select directions or suggest questions
     that are outside your persona and
     aspects/interests**
- **DO NOT suggest questions you have already asked
     before or that are similar to the questions
     you have already asked (check your history
     above)**
- Only focus on areas that align with your specific
     expertise, interests, and concerns as
     described in your persona
- If all current directions seem unrelated to your
     interests, it's better to select none and
     suggest relevant alternatives

Focus on quality over quantity – select only the
     directions that truly matter most to you and
     align with your expertise. You should refrain
     from suggesting new follow-up questions unless
     something critical is missing and directly
     relates to your interests.

You should at most suggest 1 new follow-up question.

The probability of you suggesting a new follow-up
     question is 50%.

Your response should reflect how this person would
     actually communicate when discussing their
     research preferences.

Return your response as a JSON object with the
     following structure:
\{
  "selected_directions": [
    \{
      "number": 1,
      "direction": "direction name from the
    proposal",
      "reasoning": "why this direction is most
    important to you and aligns with your
    interests"
    \}
  ],
  "new_follow_up_questions": [
    \{
      "follow_up_question": "suggested new
    follow-up question. Most of the time you
    should not suggest new follow-up questions.
    But only if you feel there's a very important
    direction missing from the proposal, suggest
    one new follow-up question at most",
      "reasoning": "why this follow-up question is
    important, missing, and relevant to your
    interests"
    \}
  ],
  "user_response": "natural response as this user
    would speak (in the format: selected numbers
    with reasoning in parentheses, then 'New
    follow-up questions:' if any)",
  "additional_context": "any additional preferences
    or clarifications related to your expertise"
\}
```

## Pause Agent

```
System Prompt
You are an expert research assistant specialized in
    making optimal pause decisions during deep
    research. Your task is to analyze the current
    research state and decide whether it's a good
    time to pause and ask for user guidance on
    which research branches to pursue, or to
    proceed with the current research plan.

User Prompt
You need to decide whether to pause and ask for
    user guidance or proceed with the current
    research plan.

# Context:
**Original Query:** \{original_query\}

**Current Research Goal:** \{current_research_goal\}

**User Persona:** \{persona_text\}

**User Interests (Checklist):**
\{checklist_items\}

**Current Search Depth:** \{current_depth\} /
    \{max_depth\}

# Available Research Branches:
{branch_summaries}

# Decision Criteria:
Consider pausing (PAUSEASK) when:
- User input would help prioritize which direction
    to pursue
- There's uncertainty about which direction aligns
    best with the user's specific interests
Analyze the situation and make your decision. Your
    reasoning should be specific to the current
    research context, user persona, and branch
    characteristics.

Respond in the following JSON format:
\{
  "type": "object",
  "properties": \{
    "action": \{
      "type": "string",
      "enum": ["PROCEED", "PAUSEASK"],
      "description": "Decision to proceed with
    research or pause to ask user for guidance"
    \},
    "reasoning": \{
      "type": "string",
      "description": "Detailed explanation of the
    decision based on research context and user
    persona"
    \}
  \},
  "required": ["action", "reasoning"]
\}
```

## What You'll Do in Each Item

### Step 0 – Read the persona card

You'll see:
The **query**.
A short **persona description**.
The persona's **interested aspects** — the specific key information they expect in a high-quality report for this query.

> **Tip:** The *interested aspects* are the most important and concise part. You don't need to read every word of the full persona text, but **do** read the aspects list carefully.

---

### Step 1 – Rate aspect coverage for each report

Skim both full reports (embedded PDFs) and for each **aspect**, rate how well the report covers it:
**0** = not covered
**1** = somewhat covered
**2** = fully covered
If you choose **1** or **2**, copy-paste a short quote from that report (1–2 sentences or a short paragraph) that supports your rating.

> **Tip: You don't need to read every word**. To find evidence quickly, you can:
>
>   1. Scan headings for relevant sections.
>   2. If you'd like, open the report in a new tab and use **Ctrl/Cmd-F** to search keywords from the aspect.

---

### Step 2 – Findability

For each report, rate how easy it was to find content covering the aspects:
**0** = difficult
**1** = medium
**2** = easy
Base your rating on how easy it was for you to complete **Step 1** for each report.

---

### Step 3 – Compare two reports (A vs B)

Based on your ratings in the previous steps, pick a winner for:
**Alignment** – Which report better serves the persona's aspects?
**Focus** – How much of the report stays on the persona's aspects vs. irrelevant information?

> **Tip: You don't need to read every word**. A quick scan of section titles plus the first 1–2 sentences of each section is usually enough to judge. Some reports have a table of contents that can help you navigate, but **don't judge a report only by whether it has one**.

---

## Important

1. You must finish **all steps** for a item before moving to the next one.
2. Your progress is shown at the top of the page.
3. Judge **from the persona's perspective**, not your personal preferences.
4. We trust your judgment — do your best.

*Figure 9.* User study instructions.

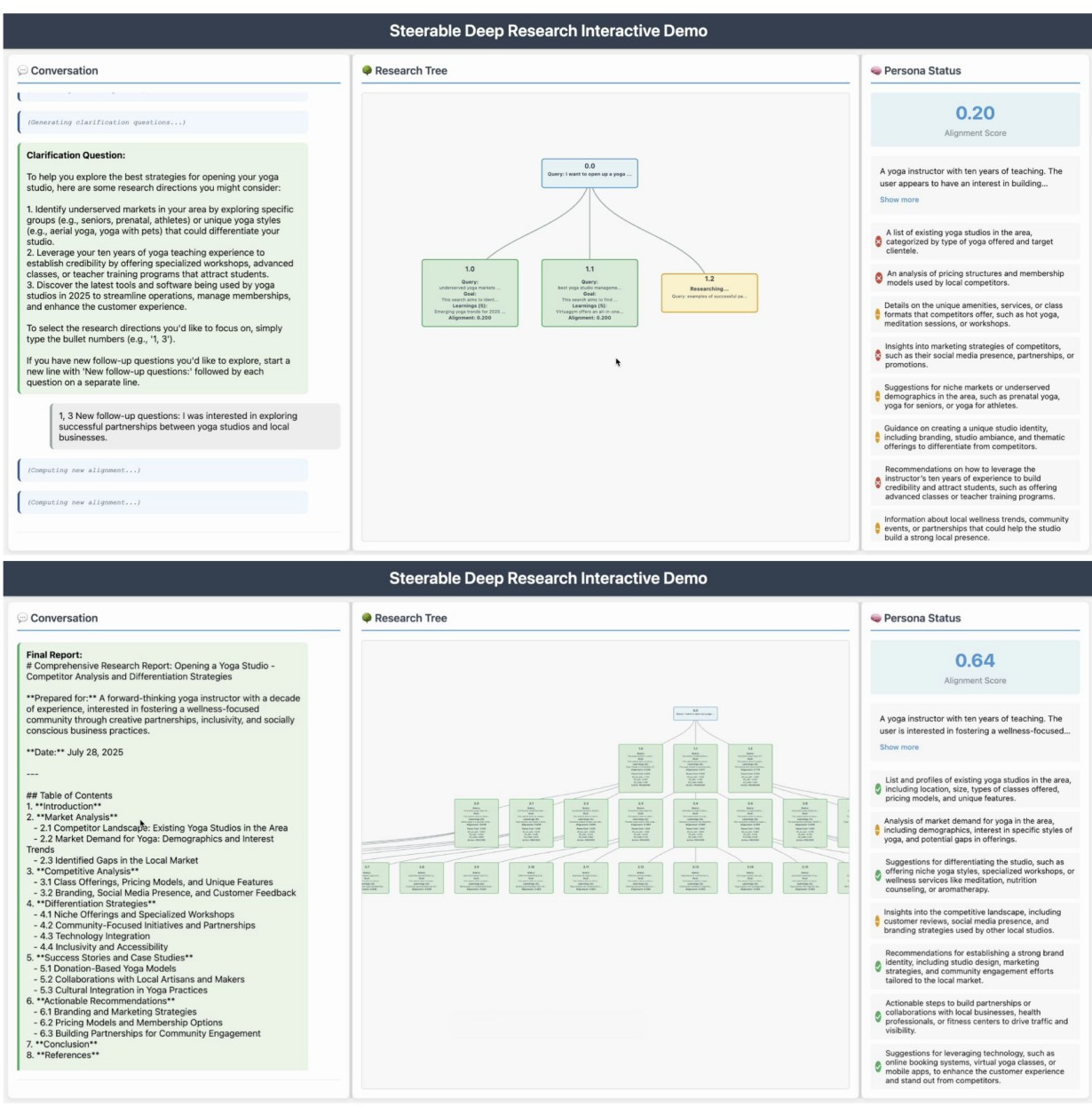

*Figure 10.* Interface of **STEER** web application.

