# OpenReview forum: "An Interactive Paradigm for Deep Research"
_ICML.cc/2026/Conference — ICML 2026 regular_

### Official Review · Reviewer_ivJR · 2026-03-10

**Soundness:** 2
**Presentation:** 3
**Significance:** 3
**Originality:** 3
**Overall Recommendation:** 4
**Confidence:** 4

**Summary:**

This paper presents STEER, a framework designed to bring mid-process, steerable control to long-horizon deep research workflows in large language models. While traditional systems often rely on a single upfront scoping step, STEER implements a cost-benefit mechanism to dynamically decide when to pause for user guidance or continue autonomously at various decision points. To maintain high alignment and exploration, the framework incorporates diversity-aware planning via Maximal Marginal Relevance (MMR) and maintains a live persona model that evolves based on real-time user interactions. Experimental results demonstrate that STEER significantly outperforms both open-source and proprietary baselines, achieving up to a 22.80% gain in user alignment metrics.

**Compliance With Llm Reviewing Policy:**

Affirmed.

**Key Questions For Authors:**

In Figure 3, does the value of C_0 hyperparameter tune on a held-out validation set?

**Limitations:**

yes

**Strengths And Weaknesses:**

Strengths:

Novel Interactive Paradigm for Deep research. STEER is the first framework to jointly integrate mid-process steering, an adaptive pause policy, and live persona modeling, addressing a major limitation in current autonomous research agents.

Interpretable Control Mechanisms: The system provides clear "knobs," such as the base pause cost ($C_0$) and tolerance budget ($Tol$), allowing developers or users to precisely balance report quality against the burden of interruptions.

Contribution of New Benchmarks: The release of a persona-query evaluation suite and a reusable data generation pipeline provides a valuable resource for the NLP community to standardize the evaluation of interactive agents.

Weaknesses:

The experimental protocol lacks sufficient clarity regarding hyperparameter tuning, particularly for the base pause cost C_0. Section 3.3 suggests that the authors fix Tol = 3, sweep C_0, and select C_0 = 0.7 because it gives a practical sweet spot on alignment and focus (Figure 3), after which the main results in Table 2 are reported at that setting. However, the manuscript does not state whether this choice was made on a held-out validation set. If C_0 was selected based on the same benchmark used for final reporting, this would amount to test-set tuning and weakens the credibility of the empirical comparison.

Some parts of the introduction and motivation could be more clearly connected. The overall motivation is reasonable, but several transitions in the introduction feel compressed. In particular, the contrast with prior systems that focus on “local clarification” or “static visualization” is only briefly stated and not fully unpacked, so the distinction from prior interactive reasoning work may not be immediately clear to readers. In addition, the contribution of the persona–query evaluation suite is introduced rather abruptly in the contribution list, with limited prior motivation in the introduction for why a new benchmark resource is necessary beyond the existing discussion of deep-research benchmarks.

---

> ### Author Rebuttal · Authors · 2026-03-28
>
> We thank the reviewer for the thoughtful and encouraging review. We are glad the reviewer highlighted the novelty of the interactive paradigm, the interpretability of the control knobs, and the value of the benchmark resource.
>
> ***1. Clarification on C0 selection (W1 + Q)***
> We view C0 differently from a conventional model hyperparameter. C0 is an **interpretable control knob** controlling interruption aversion, and Figure 3 is intended to expose the trade-off frontier between alignment/focus and pause burden, rather than to perform hidden benchmark optimization. In that sense, the sweep itself is a main result, and C0 = 0.7 is reported as a practical operating point on this transparent frontier, not as a learned parameter tuned to maximize held-out benchmark performance. This is a novel affordance of SteER: such knobs can be exposed directly to users, for example, as a slider between “Auto Pilot” and “Manual Approval”, so they can choose how much steering they want for a given query. If the goal were to prescribe one default deployment setting, a separate validation split would be cleaner; for this paper, however, we believe the exposed sweep and interpretable trade-off are the more important evidence.
>
> ***2. Clarifying the distinction from prior interactive work***
> Thank you for this very helpful feedback. We agree that the introduction currently states this distinction too compactly. By **local clarification**, we mean work where the model asks for missing information to resolve an immediate ambiguity in the current request, for example, clarification-QA pipelines [1, 2] that detect ambiguity and generate a clarification question, or ask-vs-answer methods that decide whether the current turn should be answered directly or clarified first. These methods are valuable, but they address a **local ambiguity bottleneck** in the current turn, rather than an end-to-end policy for **where to pause in a long-horizon research trajectory, how pausing should trade off with exploration, and how user goals should be updated mid-process**. That broader control problem is what SteER is designed to address.
> By **static visualization**, we mean systems that expose reasoning traces for inspection, editing, or analysis. For example, *Interactive Reasoning* [3] visualizes chain-of-thought as a hierarchy of topics and allows user modification, while *ReasonGraph* [4] provides a platform for visualizing and analyzing sequential or tree-based reasoning paths. These tools improve transparency and human oversight, but they do not implement an **adaptive pause mechanism** or **live persona modeling** that conditions planning, branch utility, and synthesis within a single long-horizon research tree. Table 1 and Appendix A were intended to make this capability distinction explicit. We appreciate the reviewer flagging that we did not unpack this sufficiently in the introduction. Due to space constraints, we kept it brief there, and if accepted, we will use the additional page to move this explanation forward and make the distinction much clearer.
>
> ***3. Motivation for the persona–query evaluation suite***
> We appreciate this question. Existing deep-research benchmarks are valuable for standardizing **research-style queries** and long-form report evaluation, but they do not provide the **persona-conditioned, aspect-grounded targets** needed to evaluate alignment and focus for interactive, user-steerable research agents. This is exactly the gap our persona–query suite is designed to fill: it pairs deep-research-worthy queries with query-conditioned personas and **actionable, measurable aspect checklists**, enabling controlled evaluation of persona alignment under interaction. Due to space constraints, we kept this motivation brief in the current draft; if accepted, we will make this need and positioning much clearer in the introduction.
>
> Thank you again for these comments; they sharpen the paper’s framing. We hope the main takeaway remains clear: SteER introduces an interpretable interactive paradigm for deep research, improving persona alignment and report focus over strong baselines.
>
> ***References:***
> [1] Andukuri, Chinmaya, et al. "STaR-GATE: Teaching Language Models to Ask Clarifying Questions." First Conference on Language Modeling.
> [2] Zhang, Michael JQ, W. Bradley Knox, and Eunsol Choi. "Modeling Future Conversation Turns to Teach LLMs to Ask Clarifying Questions." The Thirteenth International Conference on Learning Representations.
> [3] Pang, Rock Yuren, et al. "Interactive reasoning: Visualizing and controlling chain-of-thought reasoning in large language models." arXiv preprint arXiv:2506.23678 (2025).
> [4] Li, Zongqian, Ehsan Shareghi, and Nigel Collier. "ReasonGraph: Visualization of reasoning methods and extended inference paths." Proceedings of the 63rd Annual Meeting of the Association for Computational Linguistics (Volume 3: System Demonstrations). 2025.

---

> > ### Author Rebuttal · Reviewer_ivJR · 2026-04-03
> >
> > Although the rebuttal promises to improve the clarity, I retain my score of week accept as I need to take the clarity of the original submission into consideration

---

### Official Review · Reviewer_Gv8F · 2026-03-11

**Soundness:** 2
**Presentation:** 3
**Significance:** 3
**Originality:** 2
**Overall Recommendation:** 4
**Confidence:** 3

**Summary:**

In this paper, the authors introduce SteER, a framework for steerable deep research that allows user intervention during the automated search process. It uses a cost-benefit analysis to decide when to pause for feedback and employs a live persona model to align the final report with evolving user preferences.

**Compliance With Llm Reviewing Policy:**

Affirmed.

**Final Justification:**

The authors addressed the majority of my concerns, like the type of queries, the resource consumption, thus I increased my ratting accordingly.

**Key Questions For Authors:**

1. How does the performance vary across different types of queries, such as simple fact-finding tasks that might trigger earlier pauses versus complex, exploratory research?
2. Can the authors provide specific data on resource consumption, such as API costs or time costed, to support the efficiency claims?
3. How robust is the live persona model when faced with inconsistent or ambiguous feedback from a real human user?
4. Would like to see the an user study w.r.t the end to end interactive deep research process.

**Limitations:**

yes

**Strengths And Weaknesses:**

**Strength**

1. Deep research is traditionally time-consuming; an interactive paradigm can save time and improve the overall user experience.
2. The live persona model dynamically adjusts to user feedback throughout the entire research trajectory.

**Weakness**

1. While cost saving is a major claim, the experiment lacks a quantitative evaluation of actual time or token savings compared to autonomous baselines.

2. Significant portions of the dataset, including personas and user interactions, are synthetic, which makes it difficult to assess real-world performance.

3. The user study relies on static pairwise comparisons of final reports rather than an end-to-end interactive evaluation where participants actually operate the system.

4. There is a lack of thorough ablation studies specifically isolating the independent impact of the pause mechanism versus the persona modeling components.

---

> ### Author Rebuttal · Authors · 2026-03-28
>
> We thank the reviewer for the thoughtful review. We are encouraged that the reviewer sees the value of an interactive deep-research paradigm and of the live persona component. We address the main concerns below.
>
> ***1. Efficiency analysis (W1 + Q2)***
> Appendix J already notes that we do not provide a systematic wall-clock or API-cost benchmark, so we do not claim a per-run latency gains. A single runtime number for SteER mixes user-response time, pause-induced tree variation, and tool/runtime noise. What we do show is a better **interaction-efficiency** frontier: the LLM-based PauseAgent far exceeds the Tol = 3 budget, while SteER with C0 >= 0.4 stays within budget with fewer than 3 pauses on average, and higher C0 yields fewer but more impactful interventions. Our pause rule also includes an execution-cost term, a proxy for latency and spend. We therefore view the paper as establishing a better alignment–user-burden frontier and a concrete mechanism for reducing wasted effort from misaligned monolithic runs, while a full runtime/API-cost study remains future work.
>
> ***2. Synthetic data/interactions (W2 + Q3)***
> The simulator is a deliberate choice for **controlled, reproducible system-level comparison** on identical persona-query pairs, which is difficult with free-form live users. We use the User Agent as a scalable diagnostic tool, not a replacement for human validation; it has very high precision, and its recall closely tracks final-report alignment. We also include a human study on final reports, where SteER is preferred on Alignment and Focus and improves Coverage and Findability. A full live interactive study would be valuable future work, but we view it as complementary to the controlled evaluation here.
>
> ***3. End-to-end interactive user study (W3 + Q4)***
> The current human study validates final report quality under controlled conditions. The paper already identifies end-to-end interactive user studies, measuring task success, time to insight, perceived control/trust, and cognitive load, as a key next step. Such studies would complement our controlled design by directly measuring real-user benefit, but are beyond the scale of this initial work. We nonetheless believe that the level of detail in this work, together with the encouraging results, will help spur further research on steerable long-running agents.
>
> ***4. Pause mechanism and persona modeling ablation (W4)***
> We believe these are not simple orthogonal on/off switches. A no-pause SteER variant is conceptually the autonomous end of the spectrum, though it is not literally identical to the evaluated autonomous baselines because it would still retain SteER’s own diversified planning, branch-utility scoring, and automatic expansion logic. In that sense, our current analysis already goes beyond a binary pause/no-pause ablation: by sweeping C0, **we effectively scan the autonomy–interaction spectrum**.
> Conversely, a w/o persona modeling variant changes the objective of the system rather than removing an auxiliary module. In particular, the alignment-gain term is defined relative to the current inferred aspect set; without live persona modeling, that term effectively disappears. The system would still optimize generic signals such as Explore, InfoGain, and execution cost, but **it would no longer optimize toward persona alignment** at all.
>
> ***5. Query types and ambiguous feedback (Q1 + Q3)***
> These are both important and exciting directions, but they sit outside the core scope of the current paper. For query type, our benchmark is intentionally built on deep-research-worthy queries. Accordingly, **we do not view simple fact-finding as the target setting for evaluating a deep-research framework itself**; in practice, such cases are better handled by a higher-level **routing** or **early-stopping** mechanism that decides whether a full deep-research workflow is needed. This matters in production, especially since users do sometimes invoke deep-research tools for simpler questions, but we see it as a deployment extension rather than something that diminishes our core contribution: interactive control for long-horizon research tasks.
>
> Ambiguous or inconsistent feedback is a highly realistic and interesting challenge. In this paper, the User Agent is designed for high fidelity to the target persona, so feedback stays consistent with the ground-truth aspect set and enables controlled evaluation. At the same time, we believe the **interactive nature of SteER is exactly what opens the door to studying users who revise their goals mid-process**. The current live-persona module already handles *new* preferences by updating the inferred persona from user feedback. A natural next step is to extend this to handling *conflicting* preferences. We see this as an exciting direction, closely tied to agent memory conflict handling and preference tracking, that builds on rather than detracts from SteER’s core contribution.

---

> > ### Author Rebuttal · Reviewer_Gv8F · 2026-04-03
> >
> > The authors addressed the majority of my concerns, I will increase my score accordingly.

---

### Official Review · Reviewer_4T75 · 2026-03-13

**Soundness:** 2
**Presentation:** 3
**Significance:** 3
**Originality:** 3
**Overall Recommendation:** 4
**Confidence:** 3

**Summary:**

The paper introduces STEER, a framework for deep research agents that aims to improve alignment by allowing the system to decide whether to pause and ask for user input or to continue autonomously. The method includes 3 main parts: (1) diversity-aware branch proposal, (2) a pause/proceed decision rule, and (3) a live persona model. The evaluation is done on a synthetic persona-query benchmark derived from the benchmark Researchy Questions. The paper reports improvements over both open-source and proprietary baselines. The paper also includes a small meta-evaluation for the LLM judge, reporting a moderate but statistically significant correlation with humans.

**Compliance With Llm Reviewing Policy:**

Affirmed.

**Key Questions For Authors:**

1. o4-mini-initial slightly outperforms o4-mini-full on some metrics, which again suggests that more persona information may sometimes hurt due to prompt overload, distraction, or weak utilization by the baseline. Could this suggest that the query-persona pairs might not be using minimal viable personas? Can the authors elaborate more on why we see this apparently counterintuitive result?
2. Why is UCB the right exploration bonus for this setting? Please clarify whether the choice is principled or mainly heuristic, and discuss why alternative exploration mechanisms were not considered.

**Limitations:**

yes

**Strengths And Weaknesses:**

## Strengths

- The paper studies an interesting problem. Current deep-research systems are largely organized around “plan once, then run for a long time,” whereas many real research workflows benefit from course correction after partial evidence emerges.
- The paper introduces an interesting and well-constructed system. The combination of pause decisions, live persona updates, and exploration-aware scoring is an interesting contribution at the systems level. The overall design is intuitive and the paper communicates the high-level loop clearly.
- The decomposition of the methods into separable parts is valuable as this allows for the method to be adapted for numerous use cases and also makes it more interpretable.

## Weaknesses

- The evaluation uses a fully-simulated setting. The offline evaluation uses a User Agent conditioned on the full ground-truth persona and aspects, and the system is evaluated largely by how well it performs under that simulator. The human study helps, but it is relatively small and does not evaluate the full interactive loop with real users providing live steering during execution.
- The paper does include an LLM-as-a-judge sanity check, but the reported Pearson correlation with human aspect scores is only r \= 0.34.
- The baseline choice could be stronger. The authors do not provide a justification for choosing o4-mini over stronger models such as OpenAI’s GPT-5. Moreover, it would be beneficial to also test other proprietary agents, such as Gemini, Grok and Perplexity Deep Research models.
- The method expands a research tree with fixed depth 3 and breadth 3, and the paper says outputs share the same token cap across STEER and the open-source baselines. The branching design itself can create large internal token consumption through additional planning, scoring, persona updating, and potential pauses, even if the final report budget is capped.
- The choice of depth 3 / breadth 3 feels under-motivated. The authors state that Tol matters less in shallow trees and that long-horizon settings are future work, but this also weakens the claim that the paper solves interactive control for deep research in the broader sense. Real deep-research agents often iterate more extensively than a 3x3 tree in practical deployments, and the paper does not show whether the same policy remains effective when depth and breadth vary.
- The paper uses a UCB exploration bonus. That intuition is understandable, but this is not obviously a classical bandit setting: the arms are not stationary, branch values are not repeatedly sampled in the usual way, and the system is doing structured search conditioned on an evolving persona. The authors do not justify why UCB specifically is the right inductive bias here, nor why alternative exploration bonuses or model-selection mechanisms were not considered.
- The paper ablates Explore, InfoGain, and diversity-aware exploration, and reports that it sets both lambda\_info and lambda\_exp to 0.5 for balance. However, I did not see a real sweep over these weights. It would be beneficial to have a sensitivity analysis on these parameters rather than only a binary removal.

---

> ### Author Rebuttal · Authors · 2026-03-28
>
> We thank the reviewer for the thoughtful review and are glad they found the problem important and the system design intuitive.
>
> ***1. Simulated users and human study (W1)***
> The simulator is a choice for **controlled, reproducible system-level comparison** on identical persona-query pairs, which is difficult with free-form live users. We use the User Agent as a scalable diagnostic tool, not a replacement for human validation; it has high precision, and its recall closely tracks final-report alignment. We also include a human study on final reports, where SteER is preferred on Alignment and Focus and improves Coverage and Findability. A full live interactive study would be valuable future work, but is complementary to the controlled evaluation here.
>
> ***2. LLM-as-judge correlation (W2)***
> We view r = 0.34 as supportive when interpreted in context: this is an aspect-level comparison on an inherently subjective long-form evaluation task, and the correlation is statistically significant (p < 0.0001), positive for every annotator, and reasonably consistent across annotators. More importantly, our conclusions do not rest on judge metrics alone: the human study independently validates the main claims.
>
> ***3. Stronger proprietary baselines, and o4-mini-initial vs. o4-mini-full (W3+Q1)***
> We meant to isolate SteER’s **systems contribution**, mid-process pausing plus live persona modeling, rather than run a frontier-model leaderboard. In our controlled setup, SteER and the open-source frameworks share the same base model (**GPT-4o**), fixed tree, and output budget. For proprietary systems, we prioritize models that can be evaluated **programmatically and reproducibly at scale** in a common harness. For OpenAI, this means benchmarking dedicated deep-research API models (o3-deep-research and  o4-mini-deep-research, the latter optimized for speed/efficiency) rather than a flagship general-purpose model such as GPT-5. More broadly, direct comparison to closed product stacks would confound the systems question with differences in model scale, orchestration, tooling, and product-side prompting.
>
> That o4-mini-initial slightly exceeds o4-mini-full on some metrics is consistent with our hypothesis and previous research: **more upfront persona detail can sometimes distract a one-shot system or get lost over long trajectories**. This is precisely why SteER uses live persona modeling rather than assuming that more persona text is always better. Since our personas and aspects are query-conditioned, diversity-filtered, and actionable, we do not view this as evidence that the query-persona pairs are ill-formed.
>
> ***4. Fixed tree size and internal compute (W4+W5)***
> The fixed depth/breadth is a **controlled evaluation choice**, not the intended limit of the paradigm. It enables matched, reproducible comparison before moving to dynamic planners. We do not claim to resolve interactive control for all depth/breadth regimes, or to match autonomous baselines in internal compute or latency, since SteER adds planning, persona updating, and pause overhead. What we do show is a better **interaction-efficiency frontier** in this controlled setting: SteER stays within the pause budget, averages fewer than 3 pauses, and still achieves substantial alignment gains. These gains are also not limited to fixed-tree open-source baselines: SteER outperforms o4-mini-deep-research as well, even though its internal planning depth/breadth is not exposed. We therefore view the current results as strong evidence for adaptive mid-process control; if anything, as horizons grow, the value of correcting misalignment earlier should only increase, while the exact best knob values under those broader regimes remain open.
>
> ***5. Why use UCB-style exploration bonus (W6+Q2)***
> Our use of UCB is deliberately limited. We do not treat SteER as a classical stationary bandit or claim standard UCB guarantees. Rather, Explore is a lightweight **count-based bonus over facet tags** that rewards under-explored directions and decays with reuse. This gives the inductive bias we want, avoiding repeated collapse onto the same facet, while remaining simple and interpretable. It complements the other components: MMR promotes local diversity, InfoGain semantic novelty, and Explore longer-horizon facet coverage. We will clarify that “UCB-style” should be understood as an optimism-based exploration prior.
>
> ***6. Sensitivity of lambda_info and lambda_exp (W7)***
> A fuller sweep would be informative, but the current ablation already shows clear directional sensitivity: removing Explore hurts Focus, Depth, and Breadth most, while removing InfoGain hurts Alignment most. This motivates the balanced lambda_info = lambda_exp = 0.5.
>
> Thank you again for these comments; they sharpen the paper’s framing. We hope the main takeaway remains clear: SteER introduces an interpretable interactive paradigm for deep research, improving persona alignment and report focus over strong baselines.

---

> > ### Author Rebuttal · Reviewer_4T75 · 2026-04-03
> >
> > The authors have mostly addressed my questions. I will maintain my score.

---

### Official Review · Reviewer_z2MG · 2026-03-13

**Soundness:** 2
**Presentation:** 3
**Significance:** 3
**Originality:** 3
**Overall Recommendation:** 4
**Confidence:** 4

**Summary:**

The authors consider a central concept: how to introduce interpretable, mid-process control into long-horizon deep research workflows. This research presents the concept of STEER, an interactive framework that uses a cost-benefit formulation to decide whether a research agent should pause for user input or proceed autonomously. The system actively maintains a live persona model that updates throughout the session to better align with the user's evolving intent. The authors evaluate their framework using both an LLM-simulated user agent for automated metrics and a human study, comparing STEER against open-source and proprietary baselines.

**Compliance With Llm Reviewing Policy:**

Affirmed.

**Key Questions For Authors:**

1. The human evaluation is a major problem as double-blind is almost impossible. How can you confidently claim human preference when annotators could likely identify STEER's outputs due to structural or stylistic artifacts resulting from its unique interactive process?
2. You explicitly state that you omitted a latency analysis. Given that user experience in an interactive system is dictated by speed, how much slower is a STEER end-to-end run compared to the autonomous baselines in real wall-clock time?
3. The simulated User Agent acts rationally to optimize the persona. Have you tested STEER's robustness against real-world human behavior, such as users providing contradictory mid-process feedback or abandoning the clarification process entirely?
4. How sensitive is the system's performance to the base pause cost (C0) and tolerance budget (Tol) across vastly different domains (e.g., highly technical medical queries vs. general pop-culture queries)?

**Limitations:**

No. While the authors acknowledge the use of simulated personas and the lack of runtime analysis, they completely fail to discuss the severe blinding limitations of their human study. They also do not address the fragility of their heuristic cost-benefit parameters or the potential for their token-based cost proxy to completely fail in real-world web environments. They must be far more transparent about how these factors artificially inflate their perceived success.

**Strengths And Weaknesses:**

**Strengths:**
- The paper tackles a highly relevant and practical problem: most current deep research systems rely on rigid, one-shot scoping that fails to adapt if user intent shifts.
- The proposed architecture successfully integrates several distinct components, diversity-aware exploration, cost-benefit pause decisions, and live persona modeling into a cohesive pipeline.
- The automated evaluation shows empirical gains, with the system outperforming baselines on alignment metrics.


**Weaknesses:**

- **Flawed Human Evaluation (Blinding Issues):** The human evaluation is a major problem as double-blind assessment is almost impossible. Because STEER is inherently an interactive system that asks clarification questions and adapts dynamically, its final reports and interaction traces likely contain structural artifacts or conversational tones that easily distinguish them from the static outputs of baselines like GPT-Researcher. Annotators can likely guess which system is which, severely compromising the integrity of the pairwise preference study.
- **Over-Reliance on Simulated Users:** The primary large-scale evaluation relies on a "User Agent" simulated by an LLM to provide steering signals. Simulated agents are highly predictable, hyper-rational, and do not suffer from cognitive fatigue. Real human users are noisy, provide contradictory feedback, and abandon tasks if asked too many questions.
- **Lack of Latency and Runtime Evaluation:** The authors explicitly admit they do not provide a systematic runtime or latency analysis. In an interactive system, wall-clock time and user-perceived latency are critical to usability. Pausing, maintaining a search tree, and running multiple LLM evaluations at each node likely introduces severe bottlenecks that the paper completely sweeps under the rug.
- **Heuristic and Fragile Cost-Benefit Logic:** The pause decision relies heavily on hardcoded assumptions and hyper-parameters, specifically the base pause cost (C0) and the tolerance budget (Tol). These heuristic variables likely require extensive manual tuning per user or per domain and may not generalize well to real-world deployment.
- **Oversimplified Execution Cost:** The system estimates the execution cost of a branch by using the token count of a saturated subtree as a proxy. This ignores the highly variable costs of web retrieval, parsing complex documents, and API rate limits, making the cost-benefit analysis fundamentally inaccurate.
- **Synthetic Evaluation Data:** The queries and personas used for evaluation were synthetically generated using GPT-4o. Evaluating an LLM-based system using synthetic data generated by the same class of models creates a closed loop that often inflates performance metrics and fails to represent the messy reality of actual human research queries.
- **LLM-as-a-Judge Bias:** The objective metrics rely entirely on a weak model, gpt-4o-mini as a judge. This introduces significant risks of LLM preference bias, where the judge may simply prefer the stylistic outputs of the underlying model powering STEER rather than genuinely assessing alignment.

---

> ### Author Rebuttal · Authors · 2026-03-28
>
> We thank the reviewer for the thoughtful and encouraging review. We are glad the reviewer finds the problem timely and practical, and we appreciate the opportunity to clarify these points.
>
> ***1. Human evaluation/blinding (W1+Q1)***
> We respectfully believe this concern may somewhat overstate what was exposed to annotators. In the human study, **annotators did not see interaction traces, clarification questions, or any intermediate system behaviors**. They only compared the **final reports** from two systems for the same query-persona pair, shown side by side in randomized order (Appendix H / Fig. 8). All systems produced report-style outputs, with no observed consistent structural cues that would make one system identifiable. Elements such as a table of contents appeared across systems. In addition, annotators were not familiar with our system or the baselines’ output format. We agree that perfect blinding is difficult in any open-ended generation setting, and we will state this limitation more explicitly; however, we believe our setup substantially reduces identification risk and supports a meaningful comparison of final report quality.
>
> ***2. Simulated users and synthetic evaluation data (W2+Q3)***
> We agree that simulated users are not a replacement for real users. Our goal here was to enable a **controlled, scalable, and repeatable** comparison across many persona-query pairs. We do acknowledge this limitation in Appendix J. At the same time, we also include a human study on final reports, which provides an important complementary validation. We agree that robustness to contradictory feedback, incomplete feedback, or user abandonment is important future work, and we will make this more explicit in the revision.
>
> ***3. Latency/runtime (W3+Q2)***
> Appendix J already notes that we do not provide a systematic wall-clock or API-cost benchmark, so we do not claim a per-run latency gains. A single runtime number for SteER mixes user-response time, pause-induced tree variation, and tool/runtime noise. What we do show is a better **interaction-efficiency** frontier: the LLM-based PauseAgent far exceeds the Tol = 3 budget, while SteER with C0 >= 0.4 stays within budget with fewer than 3 pauses on average, and higher C0 yields fewer but more impactful interventions. Our pause rule also includes an execution-cost term, a proxy for latency and spend. We therefore view the paper as establishing a better alignment–user-burden frontier and a concrete mechanism for reducing wasted effort from misaligned monolithic runs, while a full runtime/API-cost study remains future work.
>
> ***4. Heuristic parameters and domain sensitivity (W4+Q4)***
> Our intention was not to introduce opaque knobs requiring heavy manual tuning. Rather, C0 and Tol are designed as **interpretable control knobs**: C0 reflects interruption aversion, and Tol reflects the effective interaction budget. This is useful because different users and settings may reasonably prefer different interaction styles. This is also a novel affordance of SteER: such knobs can be exposed directly to users, for example, as a slider between “Auto Pilot” and “Manual Approval”, so they can choose how much steering they want for a given query. We have not yet run a dedicated cross-domain calibration study, and we agree this is important future work. Similarly, more adaptive versions, including data-driven calibration or learned policies, are **promising extensions built on top of the framework**, rather than evidence against the value of the paradigm itself.
>
> ***5. Execution-cost proxy (W5)***
> We agree that token count is a simplified proxy of real-world costs. As stated in Section 2.4 / Appendix D, we use token count of the remaining saturated subtree as a **consistent, model-agnostic proxy** for remaining work, and note that it correlates with latency and spend. Our goal was to provide a **first concrete instantiation** of the pause cost-benefit framework in a controlled setting. The primary contribution remains the decision framework itself. Because the cost model is modular, it can be expanded in future research to incorporate richer signals while remaining integrated into the same calibrated control mechanism.
>
> ***6. LLM-as-a-judge (W6)***
> Judge bias is an important concern, therefore, we do not rely on judge-based metrics alone. First, the automatic evaluation uses gpt-4.1-mini as the judge (not gpt-4o-mini). Second, we complement it with a human study. Third, we report a meta-evaluation in Appendix I showing a statistically **significant positive correlation between the LLM judge and human aspect-level judgments**.
>
> Once again, we appreciate these comments and the valuable future directions they highlight. We hope the central contribution remains clear: SteER introduces an interpretable interactive paradigm for deep research, with encouraging gains in persona alignment and report focus over strong baselines.

---

> > ### Author Rebuttal · Reviewer_z2MG · 2026-04-03
> >
> > The author has mostly addressed my concerns. I will keep my positive score as 4.

---

### Decision · Program_Chairs · 2026-04-30

**Decision:**

Accept (regular)

**Comment:**

This paper introduces SteER, an interactive framework for deep research that uses a cost-benefit formulation to dynamically decide whether an agent should pause for user guidance or proceed autonomously. It integrates diversity-aware planning with a live persona model to align long-horizon tasks with evolving user intents.

Reviewers agreed the paper tackles a highly relevant problem with a well-constructed system that introduces interpretable control mechanisms (Reviewers z2MG, 4T75, ivJR). However, they raised concerns about the evaluation relying heavily on an LLM-simulated user agent, synthetic data, and a lack of wall-clock time evaluation (Reviewers z2MG, 4T75, Gv8F). The authors' rebuttal clarified that the simulator enables controlled, reproducible comparisons and defended their execution-cost proxy (Author rebuttals to z2MG, Gv8F), but reviewers felt the evaluation lacked real-world interactive validation (Reviewer z2MG, 4T75, Gv8F). The authors also addressed clarity issues regarding their control knobs and distinctions from prior work (Author rebuttal to ivJR), though it is unclear how these issues will be fully tackled in the paper.

During the review process and discussions, all four reviewers ultimately recommended a "Weak Accept" due to the clarity issues and limited evaluation methodology. Therefore, I recommend Weak Accept.